# An Overview of the Process Mechanisms in the Laser Powder Directed Energy Deposition

Gabriele Piscopo 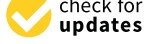, Eleonora Atzeni, Abdollah Saboori and Alessandro Salmi *

Department of Management and Production Engineering (DIGEP), Politecnico di Torino, Corso Duca degli Abruzzi 24, 10129 Torino, Italy
* Correspondence: alessandro.salmi@polito.it; Tel.: +39-011-090-7263

**Abstract:** Laser Powder Directed Energy Deposition (LP-DED) is a very powerful Additive Manufacturing process for different applications, such as repair operations and the production of functionally graded material. However, the application is still limited, and one of the main reasons is related to the lack of knowledge of the process mechanisms. Since the mechanisms involved in the process, which are mutually related to each other, directly influence the properties of the produced part, their knowledge is crucial. This paper presents a review of the LP-DED mechanisms and the relationship between the input process parameters and related outcomes. The main mechanisms of the LP-DED process, which are identified as (i) laser irradiation and material addition, (ii) melt pool generation, and (iii) subsequent solidification, are discussed in terms of input parameters, with a focus on their effects on the deposition effectiveness, and interrelation among the mechanisms of the deposition process. The results highlight the complexity of the mechanisms involved in the LP-DED process and guide engineers in navigating the challenges of the deposition process, with a specific focus on the critical parameters that should be investigated when new materials are developed, or process optimization is carried out.

**Keywords:** additive manufacturing; directed energy deposition; laser metal deposition; powder stream; melt pool; microstructure; residual stress

## 1. Introduction

Additive Manufacturing (AM) technologies emerged in the late 1980s as useful processes for producing prototypes more quickly and economically than conventional production systems [1]. The basic idea of the AM processes is that a component characterized by a complex shape can be produced directly from a three-dimensional model by adding the material layer upon layer without using expensive tools or external equipment [2,3]. Nowadays, considering processes using metal alloys as feedstock material, according to ISO/ASTM 52900:2021 [2], the main metal AM processes are Powder Bed Fusion (PBF) processes, which include Laser Powder Bed Fusion (L-PBF) and Electron Beam Powder Bed Fusion (EB-PBF), and Directed Energy Deposition (DED) processes. Figure 1 illustrates the number of metal AM systems sold from 2003 to 2021 [4]. The graph refers to all the AM systems without distinguishing the process. A sharp increase in the number of systems sold from 2016 to 2018 can be observed, and this number is expected to grow in the coming years. This trend confirms the growing interest of the manufacturing industry in the AM processes [5].

The main PBF and DED systems are listed in Table 1, showing the building volume. Nowadays, the applications of AM processes cover a wide range of industries. PBF processes are mainly used for the production of components characterized by high geometric complexity and custom design. Examples include the production of turbine blades [6,7], microwave waveguides [8,9], prostheses [10], and dental implants [11]. It is possible to observe that although PBF processes dominate the market [12], they can only be used to

produce components with a maximum size of 400 mm. DED processes, on the other hand, do not use a closed chamber, and as a result, the deposited dimensions can reach 3000 mm. Another advantage of DED processes over PBF processes is that an existing surface of a component could be used as a building platform. This feature is important for repair applications. Moreover, it is possible to change the material during the deposition process, thus obtaining components characterized by different properties in different areas. Based on these advantages, examples of the applications of DED processes are the production of large components [12,13], the repairing of high-value components [14,15], and surface remanufacturing [16,17].

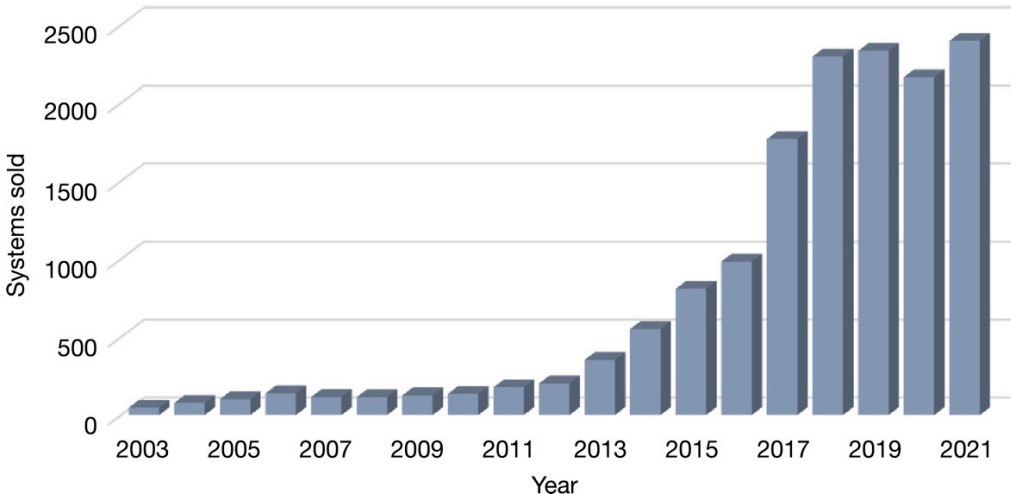

**Figure 1.** Metal Additive Manufacturing systems sold from 2003 to 2021 (adapted from [4]).

**Table 1.** Metal AM systems and their building volumes [18].

| Process Category | System | Process Name | Build Volume (mm$^3$) |
|---|---|---|---|
| Powder Bed Fusion (PBF) | GE Additive Arcam (A2x) | EBM | 200 × 200 × 380 |
| | GE Additive Arcam (Spectra H) | EBM | Φ250 × 430 |
| | GE Additive Arcam (Spectra L) | EBM | Φ350 × 430 |
| | EOS (M400) | DMLS | 400 × 400 × 400 |
| | Concept laser cusing (M2) | SLM | 300 × 350 × 300 |
| | MTT (SLM 250) | SLM | 250 × 250 × 300 |
| | Renishaw (AM 250) | SLM | 245 × 245 × 360 |
| | Realizer (SLM 250) | SLM | 250 × 250 × 220 |
| | Matsuura (Lumex Advanced 25) | SLM | 250 × Φ250 |
| Directed Energy Deposition (DED) | Prima Additive (LaserNext 2141) | DED | 4140 × 2100 ×1020 |
| | POM DMD (66R) | DMD | 3200 × 3670 × 360° |
| | Optomec (LENS 850-R) | LENS | 900 × 1500 × 900 |
| | POM DMD (66R) | DMD | 3200 × 3670 × 360° |
| | Trumpf | LD | 600 × 1000 long |
| | Sciaky (NG1) EBFFF | EBDM | 762 × 483 × 508 |

DED processes can be classified according to feedstock material and energy source [19]. Among the different DED processes, those using laser as the energy source and powder as the feedstock material are the most commonly used [20]. Different technologies that refer to the same process are developed and labeled differently, such as Laser Metal Deposition (LMD), Laser Cladding, Laser Engineering Net Shaping (LENS®), Directed Light Fabrication (DLF), and Direct Metal Deposition (DMD™). In this paper, the term Laser Powder Directed Energy Deposition (LP-DED) refers to this process.

In the LP-DED process, a deposition head delivers metal powder into the melt pool generated by a focused laser beam. Due to the localized thermal energy and the rapid move-

ment of the laser, almost instantaneous solidification of the molten material is obtained. The characteristics of the deposited part depend on a large number of parameters. According to Qi, et al. [21], up to 14 parameters can be identified: laser power, laser beam diameter, spatial distribution, shielding gas flow rate, carrier gas flow rate, travel speed, powder flow rate, powder, and building platform material properties, powder characteristics, powder feeding method, layer thickness, overlap percentage deposition strategy. These parameters define the mechanisms of the LP-DED process [22–24]. The three main mechanisms that can be identified are (i) laser irradiation and material addition, (ii) melt pool generation, and (iii) the solidification process. Moreover, complex parameter interactions are observed during the process.

An in-depth description of the LP-DED process, enabling an understanding of the physics of the process, must consider the mechanisms and parameters of the process that are closely related to each other. Figure 2 shows the three mechanisms of the LP-DED process and the relationships among mechanisms, input parameters, and outputs.

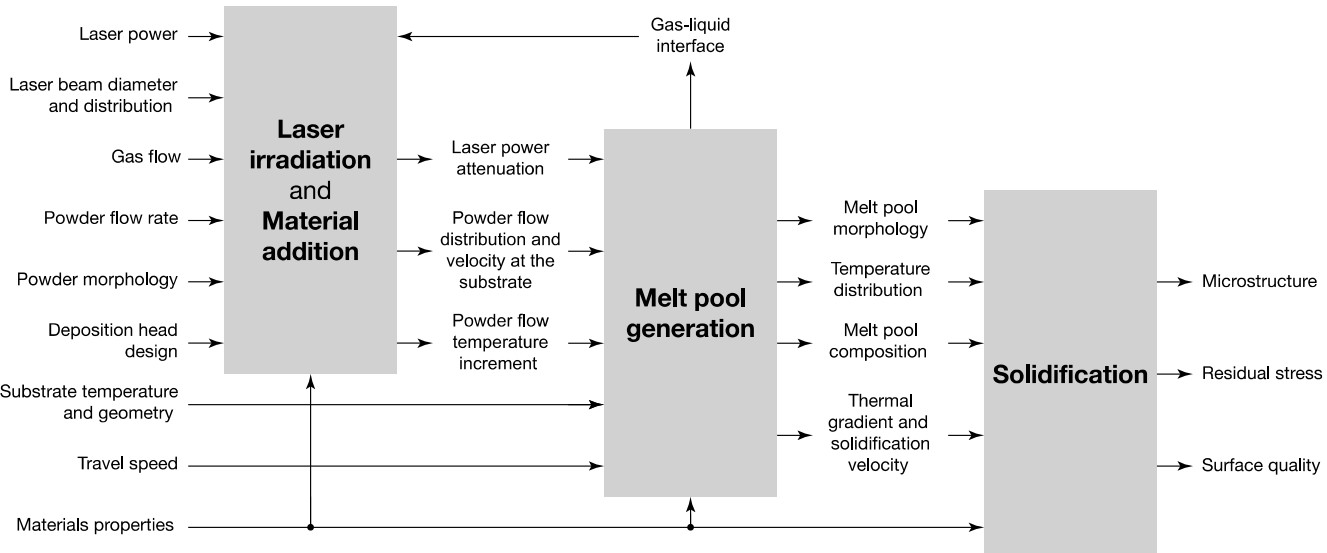

**Figure 2.** Main mechanisms and process parameters describing the physics of the LP-DED process (adapted from [23]).

However, the process is generally analyzed by considering each mechanism independently, and, as a result, the interactions among the various process parameters are neglected. Therefore, a sequential approach is used to study the process, in which the output obtained as a result of each mechanism is used as an input variable for the following mechanism. Several reviews are available in the literature, and most of them focus on the applications of the LP-DED process [12,25], modeling techniques [23], powder feeder equipment [26,27], monitoring techniques [28], and the feasibility of different alloys [29–34]. In contrast, no review in the literature focuses on a comprehensive overview of the mechanisms involved in the LP-DED process.

This work aims to describe the main mechanisms involved in the LP-DED process. At first, the laser irradiation and material addition processes are described, highlighting the main results in terms of powder distribution and laser-powder interaction. Then, the description focuses on the melt pool generation and the effect of process parameters on the temperature distribution into the melt pool. In addition, the main forces acting in the melt pool and their influence on the melt pool morphology are analyzed. Finally, the description focuses on the solidification phase with a description of the resulting microstructure, the residual stresses generation, and the analysis of the surface quality.

## 2. Laser Irradiation and Material Addition Mechanisms

The study of the powder stream process is of paramount importance as it defines the initial conditions for the generation of the melt pool [35,36]. In the LP-DED process, powders with a particle size between 30–150 µm are used [37]. Powders for AM can be produced mainly by three different technologies: Gas-Atomization (GA), Water-Atomization (WA), and Plasma Rotating Electrode Process (PREP) [38]. Powder particles produced by WA are characterized by an irregular shape elongated along a direction and dimpled surface textures [39,40]. Instead, powder particles produced by GA are characterized by a predominantly spherical shape with smooth surface textures and some satellites [40,41]. Finally, powder particles producd by PREP are characterized by an extremely spherical shape without satellites or defects [42]. Figure 3 compares the morphology of 4130 steel powder produced via GA and WA, respectively.

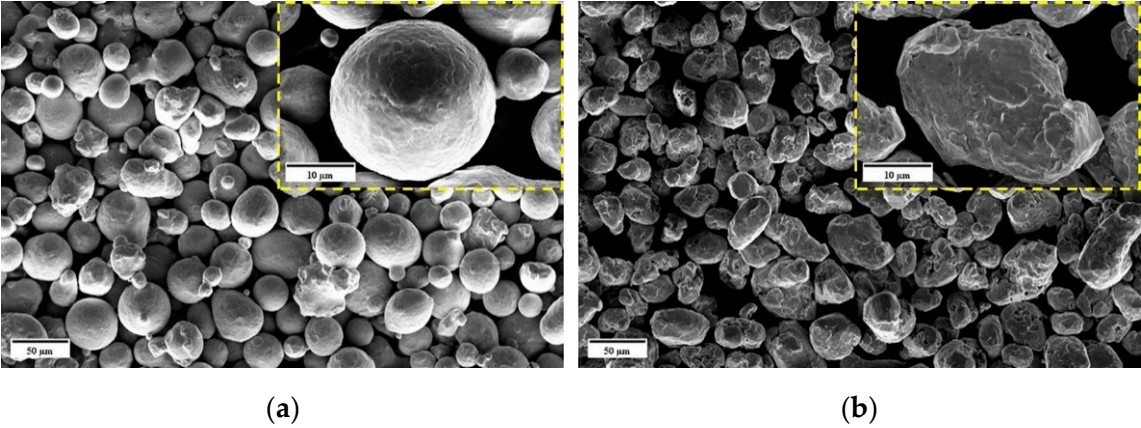

(**a**)  (**b**)

**Figure 3.** Morphology of 4130 steel powders obtained using the two systems: (**a**) gas-atomized and (**b**) water-atomized (reproduced with permission from [43]; Copyright © 2021 Elsevier).

Different experiments have been conducted to analyze the effect of powder production technologies on the characteristics of the produced parts. For instance, Pinkerton and Li [44] compared the effect of WA and GA 316L stainless steel powder on LP-DED walls. The results showed that although the use of WA powder significantly reduces the efficiency of the LP-DED process, a finer microstructure and smoother surfaces are observed. As a consequence, WA powder can be considered a viable and cheaper alternative for production [39]. Later, the same authors, during the deposition of WA and GA H13 tool steel powders, demonstrated that using GA powders resulted in higher values of deposition rate and hardness [41,45]. Zhong, et al. [46] and Ahsan, et al. [47] compared the use of GA and PREP powder particles, respectively, for the production of IN718 and Ti6Al4V samples. They revealed a higher porosity content and a lower deposition rate on the tracks produced with GA powders. Further improvements in the powder production methods could open up new opportunities for production efficiency.

As mentioned earlier, the powder particles are driven by the deposition head to produce a powder stream that interacts with the melt pool generated by the laser. Therefore, the importance of the deposition head is evident. The deposition head is mainly composed of laser optics, powder feed nozzles, shielding gas nozzles, and sensors [20]. Several deposition head configurations have been developed so far [22]. The first deposition head was the lateral one (Figure 4a), which uses a single off-axis nozzle. However, this type of deposition head has some limitations. The most important limitation is that during the deposition process with this configuration, the geometry and the characteristics of deposited tracks are direction dependent. To overcome this limitation, the coaxial configuration was introduced. Two types of coaxial deposition heads have been developed. The first uses a discrete number of symmetrically positioned nozzles (Figure 4b), while the second uses a conical nozzle (Figure 4c). The coaxial deposition head configuration is the most widely

used [37] due to its high powder capture efficiency and independence of deposited tracks from the deposition direction [48]. However, it should be considered that the lateral deposition head is more economical due to the simplicity of the equipment. Moreover, it is possible to deposit into small locations, such as inside tubes and channels [20]. Deposition head configurations are illustrated in Figure 4. Table 2 summarizes the main advantages and disadvantages of the different deposition head configurations.

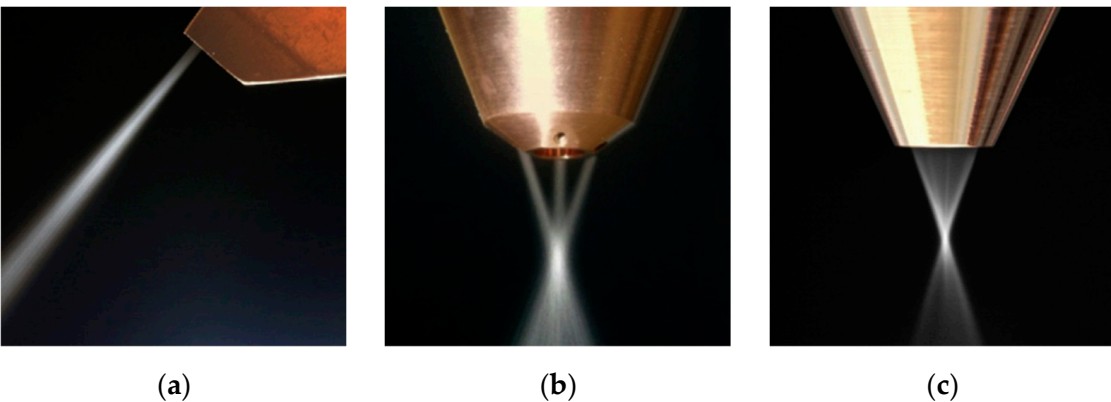

(**a**)  (**b**)  (**c**)

**Figure 4.** Deposition head configurations: (**a**) lateral configuration, (**b**) discrete coaxial configuration, and (**c**) continuous coaxial configuration (adapted from [48]).

**Table 2.** Advantages and disadvantages of deposition head configurations [48].

| Configuration | Advantages | Disadvantages |
|---|---|---|
| Lateral | Part accessibility<br>Width of deposited track: 0.5–25 mm | Directional deposition<br>Less powder efficiency<br>Alignment between powder and laser beam<br>No integrated protective gas feeding |
| Discrete coaxial | Unidirectional deposition<br>Width of deposited track: 2–7 mm<br>Applied laser power up to 5 kW<br>Unrestricted 3D functionality<br>Integrated protective gas feeding | Restricted part accessibility<br>Low powder capture efficiency<br>(diameter of the powder-gas jet in focus: minimum 2.5 mm) |
| Continuous coaxial | Unidirectional deposition<br>Width of deposited track: 0.3–5 mm<br>Applied laser power: up to 3 kW<br>Powder capture efficiency: maximum 90%<br>(diameter of the powder-gas jet in focus: minimum 400 mm)<br>Integrated protective gas feeding | Restricted part accessibility<br>Gravity influence, no deposition for tilting angles higher than 20° because of the inhomogeneous powder density distribution |

The powder stream could be analyzed considering two aspects: the powder flow and the interaction between the powder and the laser beam. The former refers to the study of the dynamics and the distribution of powder stream; the latter analyses the phenomena governing the temperature increment of the powder particles and the laser attenuation.

### 2.1. Powder Flow

In order to optimize the LP-DED process, several studies have been conducted to obtain an in-depth characterization of the powder flow. In general, it has been observed that the powder flow is influenced by particle size, powder surface morphology, and rheological properties [49]. Optimizing and understanding the powder flow could improve the capture efficiency, that is, the fraction of powder particles entering the melt pool [50]. In the LP-DED process, the capture efficiency is relatively low, less than 30%, with a huge amount of scattered powder [51]. Consequently, increasing the capture value results in

improving the overall process efficiency. Among other parameters, such as powder velocity and surface tension, the capture efficiency is mainly influenced by the distribution of the powder flow [50,52]. As a result, several studies have been conducted to understand the factors that significantly affect the powder flow.

Lin [52] performed one of the first experimental studies on coaxial flow during the LP-DED process. In his work, optical sensors and image analysis were used to measure the distribution of the powder stream. The results showed a quasi-Gaussian distribution of the powder flow in the radial direction. Later, Pinkerton and Li [53] developed a mathematical model to predict the spatial distribution of powder flow using a continuous coaxial nozzle. The results were experimentally validated using optical and image analysis techniques, which showed that it is possible to highlight two regions in the powder stream, one before merging into a single jet and another after this point. Before the merging point, the powder flow is characterized by an annular distribution, while after the merging point, a Gaussian distribution is observed. In addition, Ibarra-Medina and Pinkerton [54] showed that the powder distribution near the nozzle outlet, above the focal point of the powder stream, does not assume a perfect uniform annular distribution, but four concentration zones can be identified that correspond to the location of the nozzle inlets (Figure 5). Between the nozzle outlet and the merging point, Tabernero, et al. [55] showed the presence of a transition zone (Figure 6).

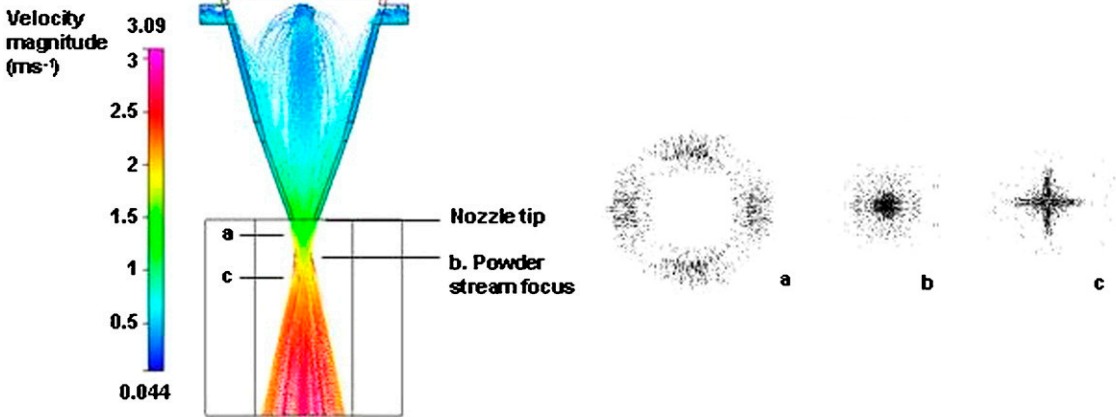

**Figure 5.** Modeled particles distribution and velocity obtained at different planes: (plane a) above the flow focal point, (plane b) at the flow focal point, and (plane c) below the flow focal point (reproduced with permission from [54]; Copyright © 2011 Taylor & Francis).

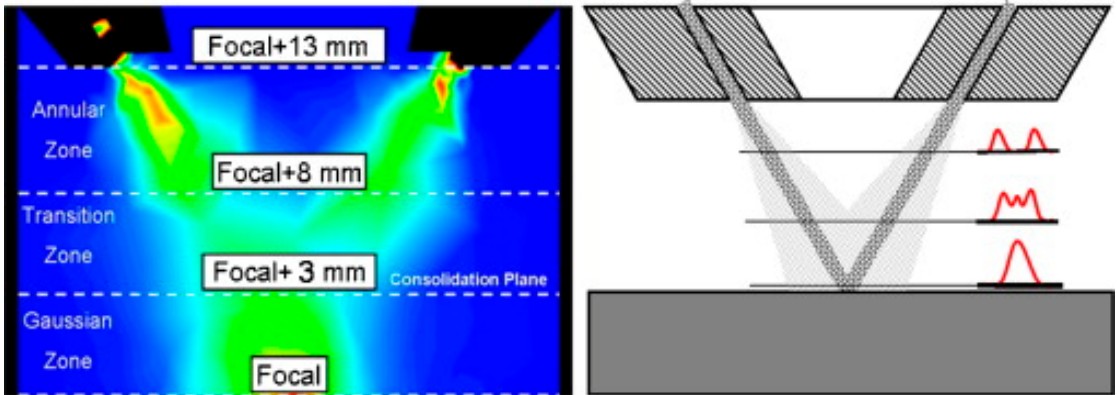

**Figure 6.** Evolution of powder flow distribution at different planes (reproduced with permission from [55]; Copyright © 2010 Elsevier).

As discussed above, the distribution of powder varies along the axial direction (z-axis). The comparison of numerical and experimental results of the powder distribution evaluated in different planes along the axial direction is shown in Figure 7. Before the merging point, the powder concentration increases with the axial distance from the nozzle exit [56]. Furthermore, the peak of powder concentration shifts close to the nozzle axis as the axial distance increases. After the focal point, the powder flow is characterized by a Gaussian distribution of powder, with a sharp decrease in peak as the axial distance increases. The point at which the maximum concentration is reached corresponds to the focal point located at some distance from the nozzle exit [57]. Focal point location and distribution strongly influence the capture efficiency and the attenuation of the laser energy. Therefore, studies focus on evaluating the effect of variables affecting distribution, such as nozzle design, powder and gas flow rate, and powder properties [58].

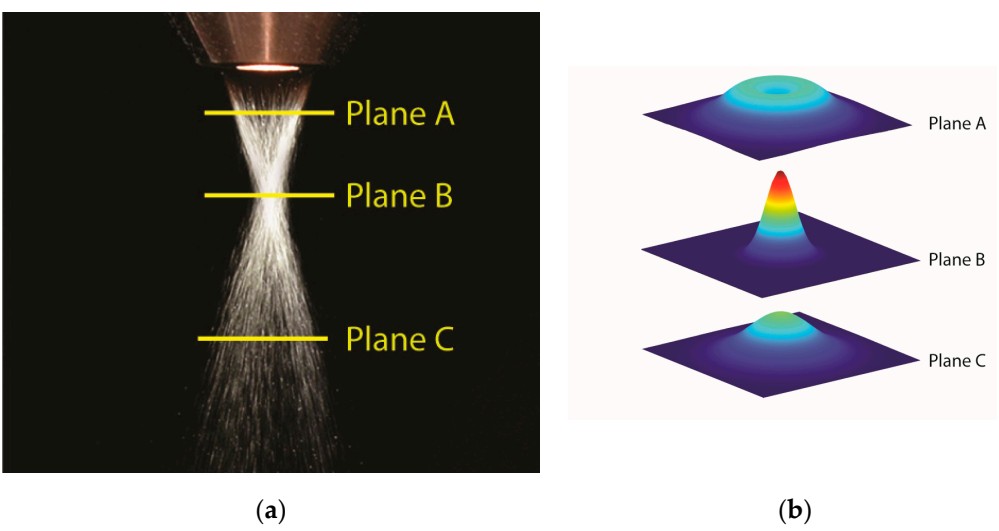

| (a) | (b) |

**Figure 7.** (**a**) Experimental and (**b**) numerical powder distribution evaluated at different planes below the nozzle exit (adapted from [53]).

In detail, the geometrical features of the nozzle that mainly influence the powder flow behavior are the nozzle configuration, the injection angle, and the outlet channel dimension [52,53,58–62]. Lin [60] developed a two-dimensional numerical model in order to analyze the effect of the deposition nozzle configuration [60]. The results showed that using a configuration with an inner nozzle positioned outward (Figure 8a), the powder concentration value increases by about 50% compared with that obtained with the inner nozzle in the inner position. In addition, the author observed that the concentration peak was strongly influenced by the velocity of the outer shielding gas, but the axial position of the powder stream focus plane remained almost unchanged.

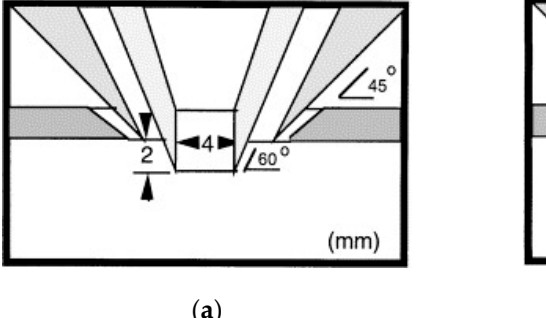 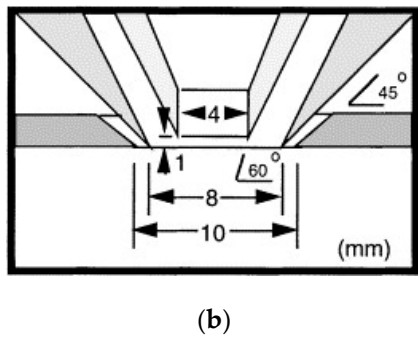

| (a) | (b) |

**Figure 8.** Coaxial deposition head configurations: (**a**) outward and (**b**) inward nozzle position (reproduced with permission from [60]; Copyright © 2000 Elsevier).

In general, it is observed that a smaller outlet size leads to a smaller powder flow diameter at the focal point [61,62]. In addition to the outlet diameter, Li, et al. [62] showed that the internal shape of the nozzles and their length also influence the powder distribution and focal point position, with the best results obtained using straight and long channels. It should also be considered that a free and undisturbed powder flow differs significantly from a flow of powder impinging on a surface [63,64]. Hence, the substrate and its distance from the deposition head play an important role [63]. The interaction between the substrate and the powder flow was studied by Ibarra-Medina and Pinkerton [63]. They found that the position of the substrate strongly influences the powder distribution. In particular, a higher concentration was obtained when the substrate was located very close to the nozzle (Figure 9).

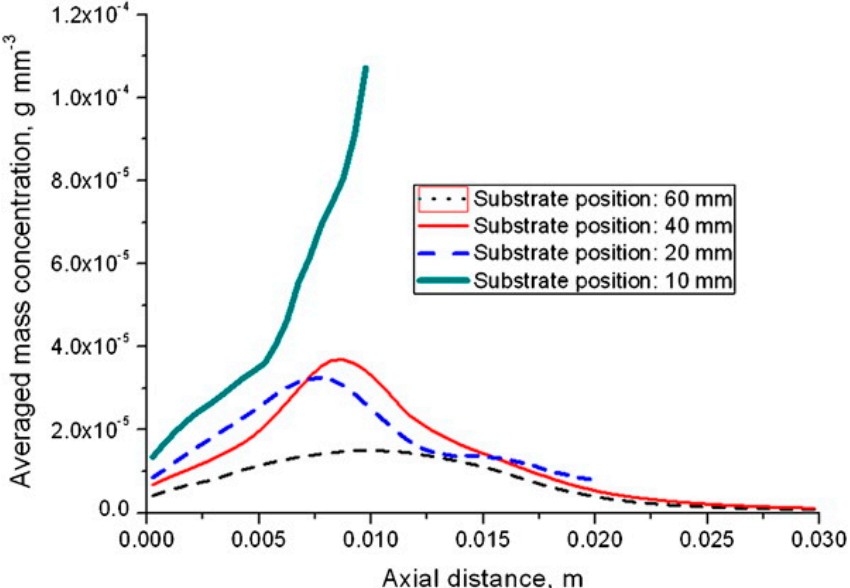

**Figure 9.** Effect of substrate position on particle concentration (reproduced with permission from [63]; Copyright © 2010 Elsevier).

Another factor that affects the distribution of the powder stream is the velocity of powder particles [65,66]. Ibarra-Medina and Pinkerton [54] showed that collisions within the nozzle walls cause a reduction in the powder particle's velocity. Then, during the in-flight time, the powders increase their momentum due to the effect of gases, and an increase in velocity is observed. Below the nozzle, the powder particle's velocity ($u_p$) slightly increases. However, a wide range of velocities was observed, with the fastest particles having twice the velocity of the slowest particles [67]. The powder particle's velocity was also influenced by the interaction between powders and laser radiation. In fact, it was observed that when the laser turned on, the particles were characterized by higher speed [68]. This acceleration was related to the light propulsion caused by the reaction between the material-vapor recoil and the irradiated part of the particle [69]. In addition, Tan, et al. [70] showed that the average powder velocity increases with increasing the shielding gas flow rate. Moreover, Kovalev, et al. [67] showed a wide velocity distribution of powder particles, as depicted in Figure 10. This wide velocity distribution was attributed to the complexity of the phenomena governing the gas-powder dynamics and the dispersion of the powder size.

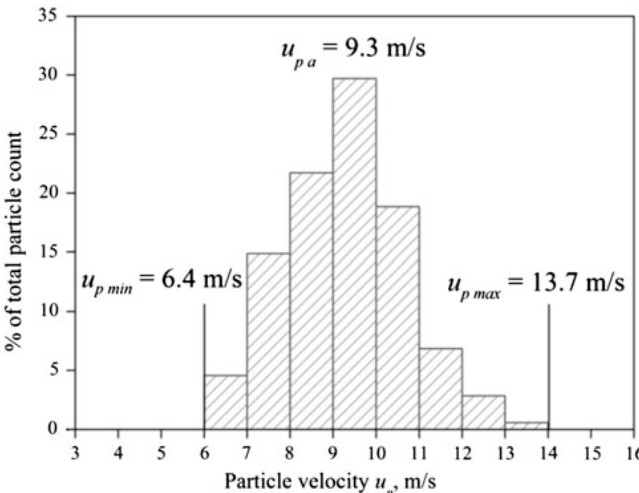

**Figure 10.** Distribution of particles velocities at a distance of 30 mm below the nozzle exit (reproduced with permission from [67]; Copyright © 2010 Springer Nature).

The relationship between the powder stream process parameters and the corresponding flow is another important aspect to consider in the optimization of the powder flow. Kovalev, et al. [67] suggested that the optimal working area, where the powder capture coefficient is maximized, is around the focal point of the powder stream. Hence, the importance of defining the powder focus plane is evident. The relationships between powder flow characteristics, such as powder distribution and powder focus plane distance, and process parameters were investigated by Liu, et al. [71] using a numerical model based on gas-solid flow. The peak of the powder stream focal point and its position were significantly affected by the velocity of the inner gas flow. Specifically, by increasing the inner gas velocity, a reduction in the peak and an increase in the axial distance of the focal point were observed (Figure 11a). Moreover, it was observed that an excessive shielding gas flow leads to defocusing of the powder flow and, as a consequence, a reduction in process efficiency [72]. The peak of powder concentration can be increased by increasing the powder flow rate [71]. It can be observed from Figure 11b that the axial distance of the focal point was unaffected by the powder flow rate. Figure 12 shows the powder flow obtained with the observation method by varying the powder flow rate.

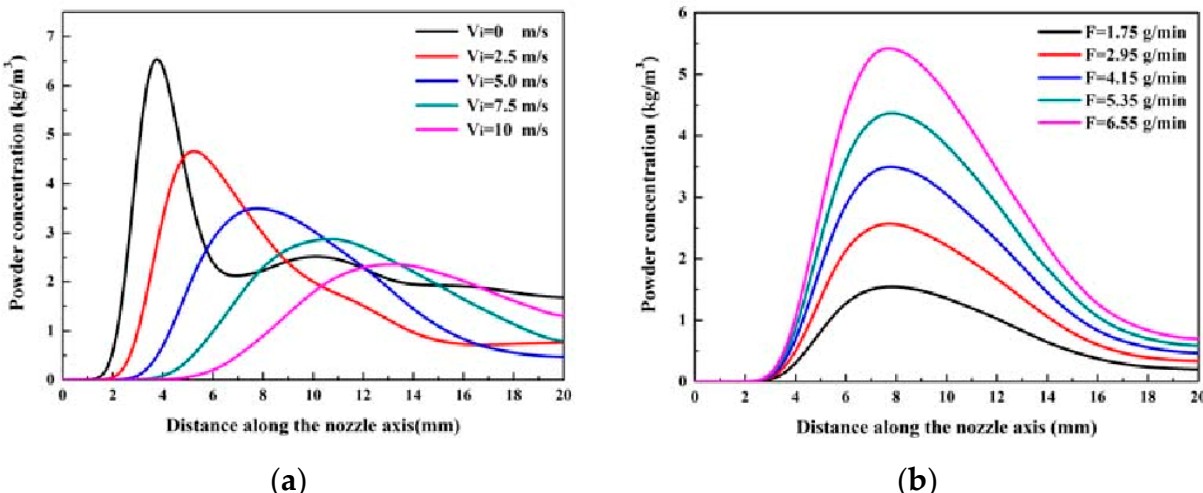

(**a**)                                                                 (**b**)

**Figure 11.** Variation of particle concentration with (**a**) inner shielding gas velocity and (**b**) powder flow rate [71].

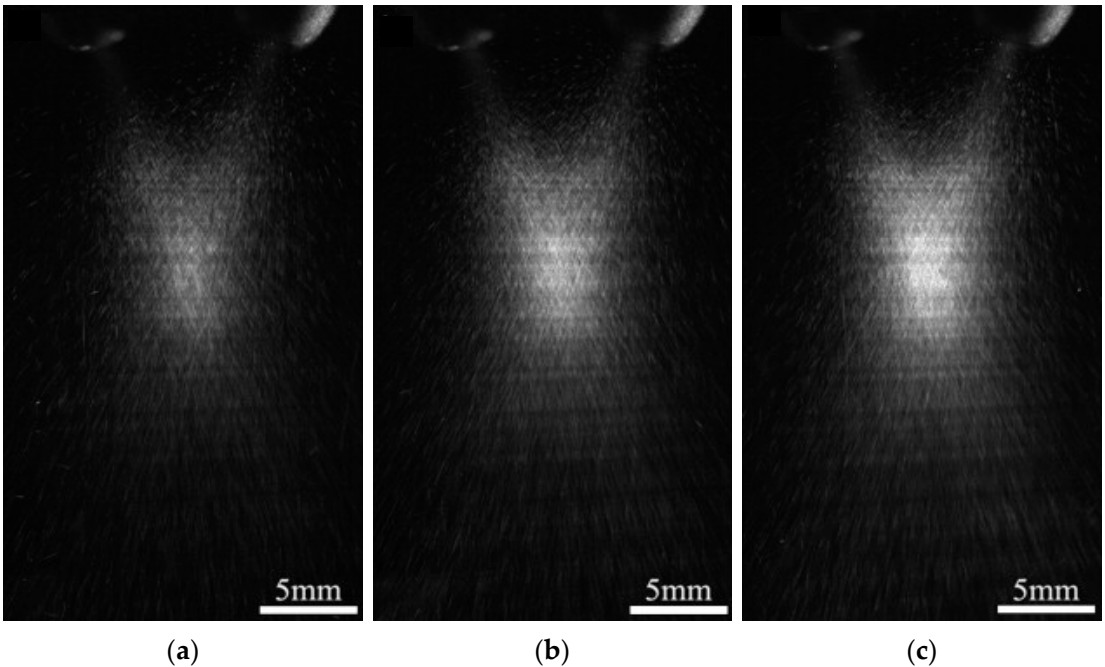

**Figure 12.** Powder distribution at different powder flow rates: (**a**) 0.052 g/s, (**b**) 0.075 g/s and (**c**) 0.088 g/s (reproduced with permission from [70]; Copyright © 2012 Elsevier).

In addition, the powder focus plane is also influenced by the density of the material [66]. Indeed, Morville, et al. [73] analyzed the particle trajectory of two different materials, 316L stainless steel and Ti6Al4V alloy. As depicted in Figure 13, both the analyzed materials exhibited a convergent trajectory of the powder stream. However, the position of the focus plane with respect to the deposition head was farther in the case of Ti6Al4V. This is because the axial effect of the gas flow is less important for heavier particles, and consequently, heavier powder particles reach the axis of the deposition head closer to the deposition head exit.

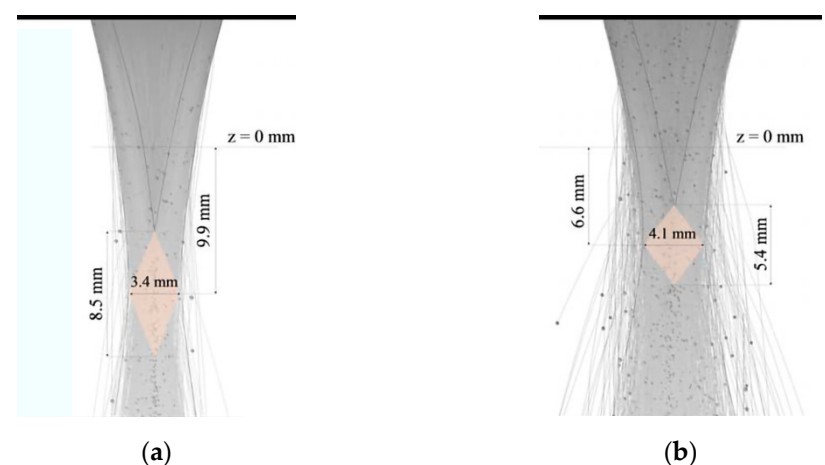

**Figure 13.** Focal plane position for (**a**) Ti6Al4V and (**b**) 316L powder (reproduced with permission from [73]; Copyright © 2012 COMSOL).

### 2.2. Powder Heating and Laser Attenuation

One of the main characteristics of the LP-DED process is the melting of the powder particles when they enter the melt pool generated by the laser. Actually, the powder particles absorb energy during the in-flight time, and their temperature increases. Depending on specific values of process parameters, powder particles can melt during the in-flight time.

However, to improve substrate adhesion and process stability, this condition is typically avoided, and process parameters are selected to melt only the substrate or previously deposited layers [20,74].

The increase in powder temperature during in-flight time is caused by the interaction between the powder and the laser beam [75]. This interaction depends mainly on the standoff distance, which is the distance between the deposition head and the substrate, and the velocity of the powder particles. The laser-powder interaction is governed by the electromagnetic and radiative properties of the powder particles and the environment. In this interaction, in addition to the rise of powder temperature, the laser power is reduced by the absorption, reflection, and scattering of light by the powder particles, resulting in the attenuation of the laser's useful power [74,76]. Figure 14 schematically illustrates the phenomena that occur during laser-powder interaction.

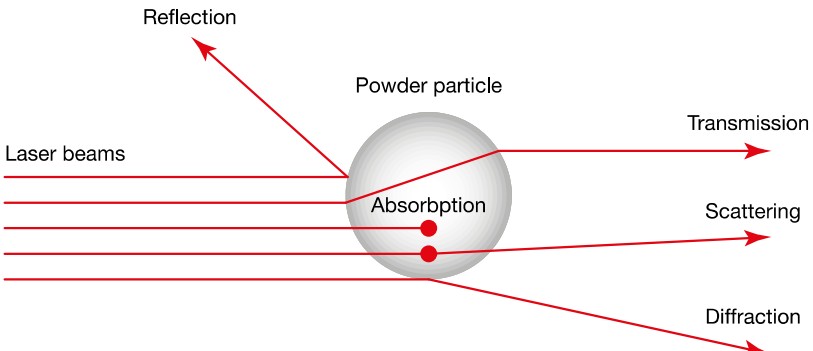

**Figure 14.** Illustration of the effects occurring during the laser-powder interaction.

One of the first analyses of powder heating was performed by Lin [77] using a mono-dimensional analytical model. He showed that the temperature of the powder highly depends on the inner gas velocity and laser power. For example, increasing the gas flow velocity from 2 m/s to 6 m/s, a variation of about 700 °C in the powder temperature was observed. The model results were compared with experimental results obtained using a pin-hole photodetector, obtaining good agreement. The experimental results showed that the temperature of the powder particles within the stream varied by about 500 °C due to the particle size distribution in the powder stream.

Transient heat transfer phenomena occur when powders interact with the laser beam. These phenomena are essentially conduction, convection, and radiation. The increase in powder temperature is usually calculated by means of an energy balance. An important parameter for studying powder heating is the Biot number [78] defined as $B_i = h_c L_p/k_p$ where $h_c$ is the convection coefficient, $L_p$ is the characteristic length (for a spherical particle with a radius of $r_p$, $L_p = r_p/3$), and $k_p$ is the thermal conductivity of the material. When $B_i \ll 1$, as in most cases during the LP-DED process, the temperature gradient within each powder particle is negligible [54,79]. Under this condition, the lumped capacitance method can be applied, and the energy balance can be described using the following equation:

$$V_p \rho_p c_p \frac{dT}{dt} = I_p \eta_p \pi r_p^2 - h_c(T - T_\infty)4\pi r_p^2 - \varepsilon_r \sigma\left(T^4 - T_\infty^4\right)4\pi r_p^2 \tag{1}$$

where $V_p$ is the particle volume, $\rho_p$ is the particle density, $c_p$ is the particle-specific heat, $T$ is the actual temperature of the particle at the time $t$, $I_p$ is the incident energy on the particle, $\eta_p$ is the particle absorptivity, $h_c$ is the convection coefficient, $\varepsilon_r$ is the radiation coefficient, $\sigma$ is the Stefan-Boltzmann constant and $T_\infty$ is the temperature of the shielding gas.

Ibarra-Medina and Pinkerton [54] analyzed the interaction between the laser beam and the powder particles. Figure 15 shows the comparison of numerical and experimental increases in powder temperature versus in-flight distance for different values of laser power.

By comparing the curves, the authors showed that the model allows the temperature increase during the process to be analyzed from a qualitative point of view.

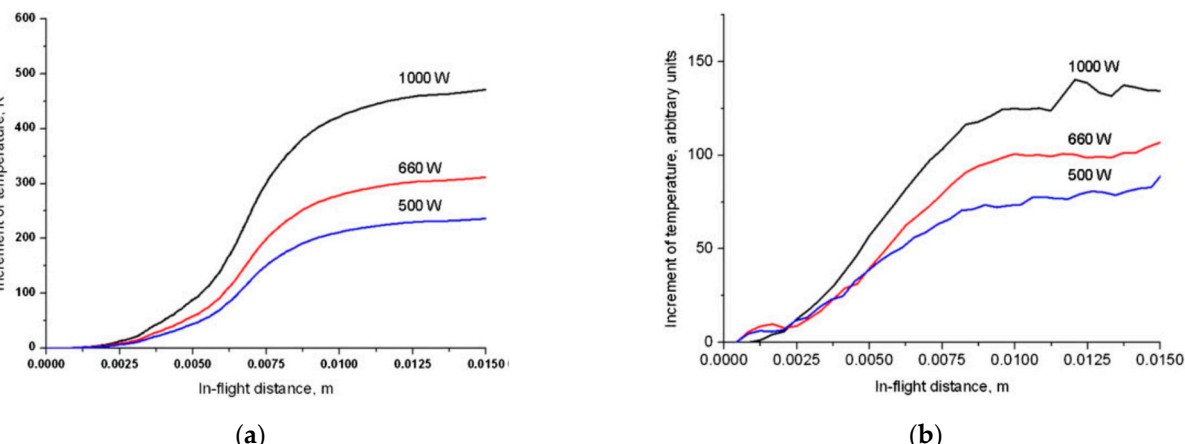

**(a)** **(b)**

**Figure 15.** (**a**) Numerical and (**b**) experimental temperature of the powder at different laser powers (reproduced with permission from [54]; Copyright © 2011 Taylor & Francis).

Several authors have successfully applied the lumped capacitance method combined with a different solution approach. Pinkerton [59] solved Equation (1) analytically and reported that the powder was characterized by different temperatures within the flow due to different trajectories. In the focus plane, the maximum temperature was reached at the centre of the laser beam, whereas a uniform temperature distribution along the radial position was observed below the focus plane.

In order to avoid the complex radiation problem involved in laser-powder interaction, several methods were used to study the power attenuation caused by the powder stream. The most used methods are (i) the Hagens-Rubens relationship, (ii) Bramson's equation, and (iii) the use of the Beer-Lambert attenuation model.

Using the Hagens-Rubens relationship, defined by Equation (2), or the Bramson equation, described by Equation (3), a time-space-temperature averaged coefficient is obtained to estimate the laser attenuation. However, it should be noted that these relationships do not consider the powder flow rate. Typical values obtained for an Nd-YAG laser are 0.36 for Inconel 718 [80] and 0.28 for 316L stainless steel [81]. The laser-surface coupling coefficient, $A(T)$, and the absorption coefficient, $\eta$, are defined by the following equation:

$$A(T) = [8\varepsilon_0 \omega \rho_e(T)]^{1/2} \tag{2}$$

$$\eta = 0.365\left(\frac{\rho_e}{\lambda}\right)^{1/2} - 0.067\left(\frac{\rho_e}{\lambda}\right) + 0.006\left(\frac{\rho_e}{\lambda}\right)^{3/2} \tag{3}$$

where $\varepsilon_0$ is the permittivity of free space, $\omega$ is the angular frequency of the laser irradiation, $\rho_e(T)$ is the material temperature-variant electrical resistivity, and $\lambda$ is the wavelength of the laser beam.

Using the Beer-Lambert attenuation model, the radial distribution of the attenuated laser beam intensity at a distance z from the nozzle can be estimated as:

$$I'(r,z) = I_0(r)exp(-\sigma_{ext}Nz) \tag{4}$$

where $I'(r,z)$ is the attenuated laser beam intensity, $I_0(r)$ is the initial laser beam intensity, $\sigma_{ext}$ is the powder extinction coefficient, and $N$ is the number of powder particles in a unit volume. $N$ is proportional to the powder flow rate [74]. He and Mazumder [82] observed that increasing the powder flow rate results in a higher value of laser beam absorption and, consequently, a lower value of useful laser power (Figure 16).

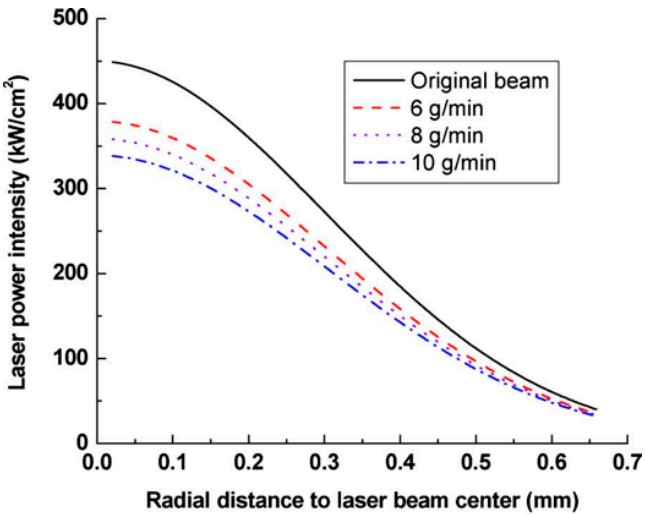

**Figure 16.** Resulting laser power intensity at different values of powder flow rate (reproduced with permission from [82]; Copyright © 2007 AIP Publishing).

The evaluation of laser absorption and, therefore, the absorption coefficient was also carried out by experimental tests. Unocic and DuPont [83] used a Thermonetics Seebeck envelope calorimeter to estimate the process efficiency during the deposition of H13 tool steel powder on a substrate of the same material and copper powder deposited on an H13 tool steel substrate. Varying the laser power between 125 and 500 W and the powder flow rate between 0.08 and 0.33 g/s, they found that the efficiency varied between 0.3 and 0.5. Using the same experimental setup, Sears [84] showed that process efficiency was about 0.42 during the deposition of Ti-6Al-4V on a Ti-6Al-4V substrate and about 0.34 depositing 316L stainless steel powder on a 304L stainless steel substrate. Peyre, et al. [85] showed that the attenuation slightly increases by increasing the powder flow rate.

Table 3 summarizes the main aspect involved in the powder flow process and the most influential factors.

**Table 3.** Summary of the factors influencing laser irradiation and material addition mechanisms.

| | Powder Flow Distribution and Velocity at the Substrate | | | Laser Power Attenuation | Powder Temperature Increment |
|---|---|---|---|---|---|
| | Govern the spatial distribution of powder particles during in-flight time | Influence the shape of the powder stream and velocity at the substrate | Influence the location of the plane of maximum powder concentration | Cause a reduction of the useful power due to laser-powder interaction | Determine the absorption of thermal energy by powder particles |
| Deposition head design | [52–55,58–62] | [61,62,68] | | [59] | |
| Stand-off distance | [56,57] | [63–68] | [57,63,64] | [54] | [54] |
| Powder morphology | | | | | [82] |
| Powder flow rate | | [67,70,71] | | [77,82,83,85] | |
| Material properties | | | [66,73] | [80,81,84] | [82] |
| Gas flow | | [67,70,71] | [71] | | |
| Laser power | | | | | [83] |

## 3. Melt Pool Generation Mechanisms

The laser power available on the substrate, i.e., that which is not attenuated by the powder flow, is focused into a small area and causes a local increase in the temperature of the building platform, generating a melt pool [86]. Figure 17a schematically illustrates the melt pool generation and the addition of material. When the powder enters the melt pool, it melts very quickly, and a raised track is obtained due to the material addition [22]. Typically, the melt pool generated during the LP-DED process is characterized by dimensions between 0.25 to 1 mm in width, 0.25 to 0.5 in height, and 0.1 to 0.5 mm in depth [20,87]. During the melt pool generation, thousands of degrees can be reached in a few milliseconds, and the values and distribution of temperature depend on several factors, such as laser power, laser

beam diameter, building platform material, and travel speed [88]. Figure 17b shows the thermal image acquired by a 12-bit digital charge-couple device camera of a moving melt pool and the corresponding temperature obtained during the deposition of a thin wall of 316L stainless steel.

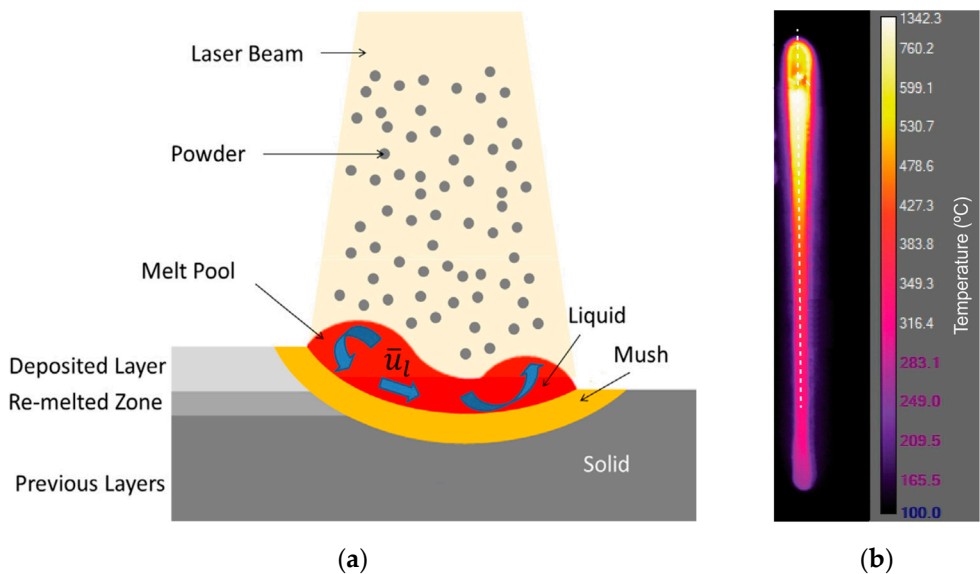

(**a**)                    (**b**)

**Figure 17.** (**a**) Schematic representation of the melt pool generation (reproduced with permission from [74]; Copyright © 2015 Elsevier) and (**b**) thermal image captured during the deposition of 316 L thin wall (reproduced with permission from [89]; Copyright © 2022 Elsevier).

Since melt pool generation is the starting point of the solid track formation, its study is of fundamental relevance to optimizing building conditions [23,74,90]. It is, therefore, important to understand which parameters have the most significant impact on the melt pool and how the melt pool dynamics affect the heat transfer. The equations governing the melt pool generation and the thermal behavior are summarized in this paragraph.

The energy available at the substrate, which generates the melt pool and overheats the molten material, can be estimated using the following equation [74,91,92]:

$$I'' = I_L - I_{loss} \tag{5}$$

where $I''$ is the useful heat flux at the substrate, $I_L$ is the heat flux at the substrate provided by the laser beam, and $I_{loss}$ is the heat loss at the melt pool-vapor interface.

Assuming a Gaussian distribution of the heat source, $I_L$ can be expressed as [74]:

$$I_L = \frac{2 \times \eta \times P}{\pi r_L^2} exp\left(\frac{-2 \times r^2}{r_L^2}\right) \tag{6}$$

where $P$ is the laser power, $r_L$ is the laser beam radius, $\eta$ is the absorption coefficient, and $r$ is the radial distance from the laser beam centre.

The heat loss at the melt pool interface includes convection, radiation, and vaporization, and it is described by the following equation:

$$I_{loss} = h_c(T(r) - T_\infty) + \varepsilon_r \sigma\left(T^4(r) - T_\infty^4\right) + \rho_L \left|\frac{\Delta r}{\Delta t}\right|_e L_v \tag{7}$$

where the first term on the right side of the equation corresponds to the heat loss due to convection, the second term corresponds to the heat loss by radiation, and the last term represents the heat loss due to the vaporization of the melt pool. Therefore, in Equation (7), $h_c$ is the convection coefficient, $T(r)$ is the temperature at the position $r$, $\varepsilon_r$ is

the emissivity, $\sigma$ is the Stefan-Boltzmann constant, $\rho_L$ is the liquid density, $L_v$ is the latent heat of vaporization and $|\Delta r / \Delta t|_e$ is the topology deformation rate due to evaporation. The topology deformation rate due to evaporation can be estimated as [92]:

$$\left| \frac{\Delta r}{\Delta t} \right|_e = c_{sound} exp \left( -\frac{\overline{L_v}}{T(r)} \right) \tag{8}$$

where $c_{sound}$ is the sound velocity in the material and $\overline{L_v}$ is the energy of evaporation per Avogadro's number.

### 3.1. Thermal Behaviour

The incident laser beam locally heats the building platform. The laser heat flux superheats the material, and the temperature obtained in the melt pool is largely above the melting temperature of the material [68]. For instance, Hofmeister, et al. [93] showed that during the deposition of a 316L thin wall, the temperature of the melt pool reached the value of about 2000 K, which is 25% higher than the melting temperature of 316L. The shape and size of the melt pool depend on the intensity distribution of the laser beam [94], the resulting temperature distribution, and the maximum temperature reached in the material. Thus, it is evident that the temperature value and its distribution are key parameters that significantly affect the melt pool. Griffith, et al. [95], during the deposition of a 316L thin wall, showed that the maximum temperature was obtained at the centre of the laser beam and then decreased linearly to the solidification temperature (1650 K) at a distance of around 2 mm from the centre, as represented in Figure 18.

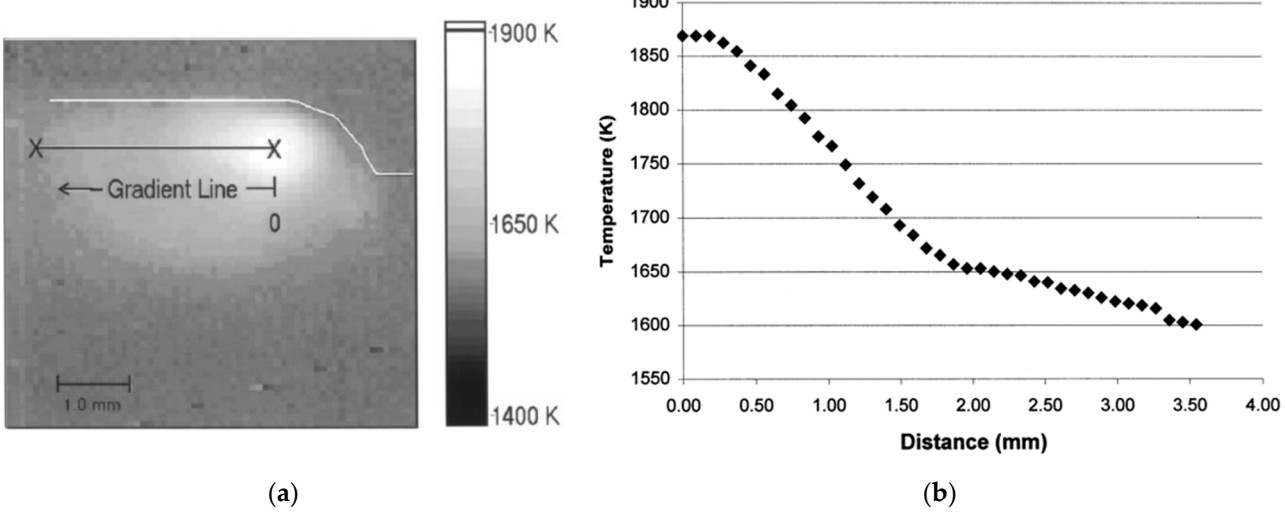

(**a**) (**b**)

**Figure 18.** (**a**) Digital image of the working area during the deposition of 316L thin wall and (**b**) temperature distribution measured across the gradient line (reproduced with permission from [95]; Copyright © 1999 Elsevier).

Among other process parameters, it is currently well-established that the process parameters that most influence the temperature distribution and, consequently, the melt pool dimensions [23] are the laser power ($P$), the powder flow rate ($Q$), and the travel speed ($v$) [20,23,96]. The following paragraphs describe the effect of these parameters on the width, height, and depth of the melt pool.

### 3.1.1. Effects of the Laser Power

It is generally accepted that increasing the laser power results in a higher temperature of the melt pool. Figure 19 shows the temperature distribution in the direction perpendicular to the travel speed for different values of laser power. It can be observed that, during

the deposition of Ti6Al4V alloy, increasing the laser power from 2 kW to 5 kW increased the maximum temperature from 1400 °C to 1850 °C, and, therefore, an enlarged melt pool width was obtained [85,97–102].

However, contradictory results have been obtained in the literature by analyzing the effect of laser power on the height of the melt pool. Pinkerton and Li [45] and Peyre, et al. [85] showed that the layer height remains constant by varying laser power values. Srivastava, et al. [99] observed that increasing the value of laser power results in a reduction of layer height. In contrast, Lee, et al. [100] and Pinkerton and Li [102] showed that layer height increases with laser power. These results indicate that different laser power levels influence the behavior of track height formation differently. In fact, the laser power strongly influences the temperature reached in the melt pool, and different laser power levels cause different convective flows in the molten material (Marangoni flows) [103]. To better understand this aspect, a more thorough investigation of the effect of laser power on melt pool dimensions should be conducted.

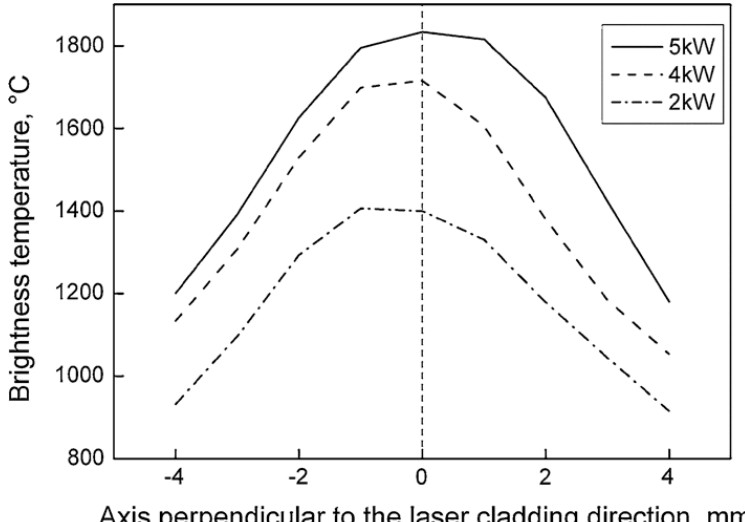

**Figure 19.** Effect of laser power on average temperature distribution into the melt pool (reproduced with permission from [104]; Copyright © 2012 Springer Nature).

### 3.1.2. Effects of the Powder Flow Rate

The powder flow rate represents the amount of additional material introduced in the deposition area. Contradictory results exist in the literature on the effect of powder flow rate on the width of the melt pool. Pinkerton and Li [45], Lee, et al. [100], and El Cheikh, et al. [105], varied the powder flow rate and demonstrated that the layer width was not influenced by this parameter. During the deposition of Ni20 powder on 316 L substrate, Hua, et al. [97] showed that the melt pool temperature decreases with increasing the powder flow rate, as depicted in Figure 20. Therefore, it was found that increasing the powder flow rate results in a smaller melt pool area [106]. The reduction of melt pool size was confirmed by Srivastava, et al. [99] during the production of Ti-48Al-2Mn-2Nb rectangular strips. The reduction in melt pool was attributed to the increase in laser attenuation due to the increase in powder flow rate. Conversely, Hu, et al. [98] and Peyre, et al. [85] observed that the melt pool width slightly increased with the powder flow rate. Consequently, the effect of powder flow rate on melt pool width depends strongly on the value of laser power and could not be studied independently [96].

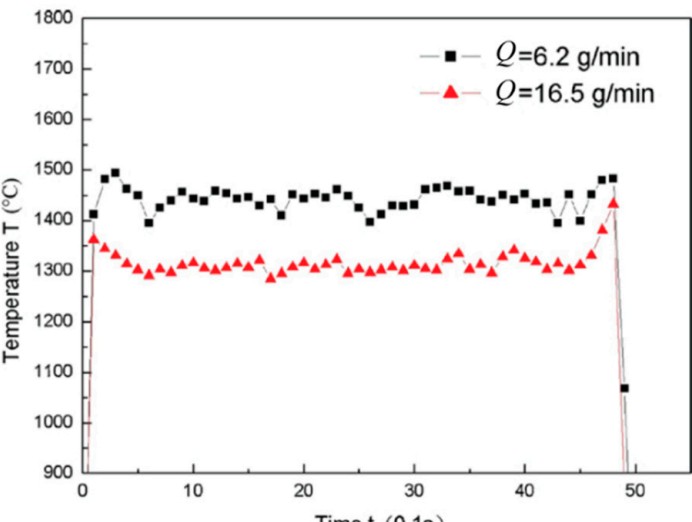

**Figure 20.** Effect of powder flow rate on temperature measurements using thermal monitoring (reproduced with permission from [97]; Copyright © 2008 Elsevier).

On the other hand, the layer height increases linearly with increasing powder flow rate [45,98,100,107–109]. Srivastava, et al. [99] observed that the layer height increases with the powder flow rate; however, after reaching an upper critical value of the powder flow rate, a reduction in layer height is observed. The existence of an upper limit of the powder flow rate that depends on the size of the melt pool should be considered. If the powder flow rate exceeds this value, the incoming particles bounce off those particles in the melt pool, creating the self-shielding effect [110].

### 3.1.3. Effects of the Travel Speed

The temperature distributions measured at the centre of the melt pool at different travel speeds are reported in Figure 21 [97]. It is possible to observe that increasing the travel speed results in a huge reduction in the temperature. Accordingly, by increasing the travel speed, a smaller width of the melt pool is expected.

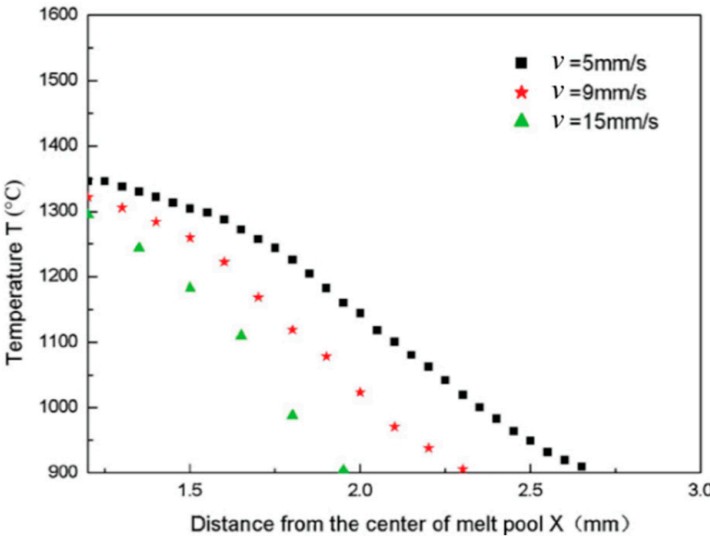

**Figure 21.** Effect of travel speed on temperature distribution along melt pool (reproduced with permission from [97]; Copyright © 2008 Elsevier).

Yellup [101] showed that for low values of laser power, the layer width decreases slightly by increasing the travel speed (Figure 22). This decrease has been confirmed by Hu, et al. [98], Wu, et al. [108], Pinkerton and Li [45], Srivastava, et al. [99], and Lee, et al. [100]. However, at a high level of laser power, the dimension of layer width is not significantly affected by the travel speed [101]. Hu, et al. [98] observed that after a critical value of travel speed, the layer width is unaffected by the variation of process parameters and is equal to the laser beam diameter. This behavior indicates that even in this case, it is not possible to consider the travel speed as an independent parameter, but it is advisable to indicate the laser power level to which it refers.

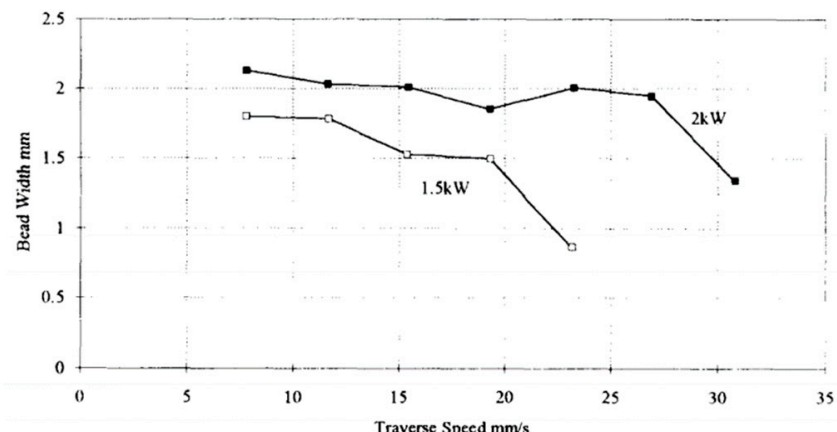

**Figure 22.** Variation of track width with respect to the travel speed varying the laser power (reproduced with permission from [101]; Copyright © 1995 Elsevier).

Hu, et al. [98], Wu, et al. [108], and Pinkerton and Li [45] observed that the layer height is inversely proportional to the travel speed. Yellup [101] and Srivastava, et al. [99] showed that before a critical value, the layer height increases by increasing the travel speed. After this critical value, a decreasing trend is observed by increasing the travel speed. The critical value depends on the material, laser power, and z-increment of the deposition head.

*3.2. Forces within the Melt Pool*

In general, it has been reported that the morphology and shape of a melt pool are mainly determined by the forces and the flows within the melt pool itself [111–113]. Two main types of forces that can be recognized are the buoyancy forces and the Marangoni forces [37,114,115]. The flow of the metal liquid caused by the buoyancy forces is illustrated in Figure 23a. Differences in material density cause the buoyancy forces due to the temperature gradient in the melt pool. Indeed, the temperature of the melt pool is higher in the centre and gradually decreases near the boundary. As a consequence, the density of the metal melt pool increases from the centre (point a) to the border (point b) [114]. The Marangoni forces cause the Marangoni flows, also named surface tension-driven or thermocapillary convection. These flows are caused by surface tension gradients. The Marangoni number (*Ma*) is used to measure the magnitude of Marangoni flows as:

$$Ma = -\frac{\partial \gamma}{\partial T} \frac{1}{\mu \kappa} t_{mp} \nabla T \tag{9}$$

where $\partial \gamma / \partial T$ is the surface tension temperature coefficient, $\mu$ is the dynamic viscosity, $\kappa$ is the thermal diffusivity, $t_{mp}$ is the melt pool thickness on the surface, and $\nabla T$ is the temperature gradient [111].

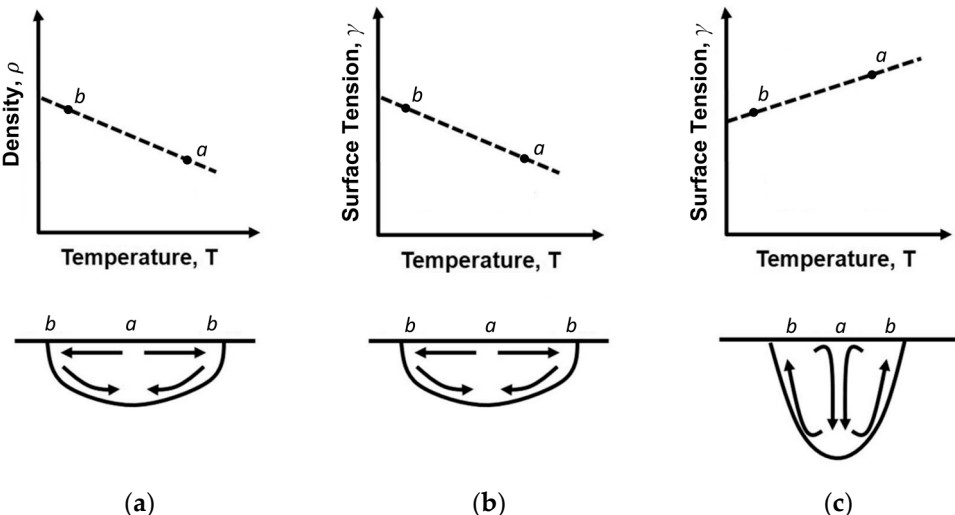

(a)  (b)  (c)

**Figure 23.** (**a**) Buoyancy forces and flows caused by the material density differences due to the temperature gradient (adapted from [114]). Marangoni flows caused by the surface tension gradients: (**b**) a negative value of surface tension temperature coefficient causes a wide and shallow melt pool morphology instead, (**c**) a positive value of surface tension temperature coefficient causes a deep and narrow melt pool morphology (adapted from [116]).

By comparing the magnitude of the flows acting in the melt pool, it is possible to observe that Marangoni flows dominate the melt pool behavior [37,111,112], which is why most works focus on this type of flow. The effect of Marangoni flows on the melt pool shape has been investigated numerically by several authors [111,115,117,118], and the effect of surface tension gradient on melt pool flows is illustrated in Figure 23. It has been shown that with a negative surface tension temperature coefficient value (Figure 23b), melt pool flows are directed from the centre to the boundary, resulting in a wide and shallow melt pool morphology. On the other hand, the melt pool becomes deep and narrow with a positive value of surface tension temperature coefficient (Figure 23c). Usually, the value of surface tension decreases with increasing temperature, so a negative value of the surface tension temperature coefficient is obtained [114]. However, the presence of surfactant elements such as sulphur and oxygen changes the value of the surface tension temperature coefficient from negative to positive [37]. Nevertheless, Marangoni flows only occur below a critical travel speed [119]. Above this critical speed, the interaction time is too low, and the surface-tension gradients are avoided. For a 316L stainless steel, the critical velocity is of the order of 90 mm/s.

The main characteristic of the melt pool generation process and the most important factors are summarized in Table 4.

**Table 4.** Summary of the factors influencing the melt pool generation mechanisms.

| | Temperature Distribution | Melt Pool Dimension | Melt Pool Morphology |
|---|---|---|---|
| | Influence the distribution and the peak of temperature into the melt pool | Determine the height, the width and the penetration depth | Determine the shape of the melt pool |
| Laser power attenuated | [88,94,104] | [45,85,97–102] | |
| Travel speed | [88,101] | [98–101,108] | |
| Powder flow distribution and velocity | [97] | [45,97,100,105–109] | |
| Material properties | [89,93] | [97] | [111–114] |
| Marangoni flows and buoyancy forces | | | [115–118] |

## 4. Solidification Mechanisms

In general, the high heating/cooling rates, pronounced temperature gradients, and bulk temperature rise involved in the LP-DED process are the key factors defining the thermal history of a component. This complex thermal history, partly produced through the LP-DED process, results in a non-equilibrium solidification which is also known as rapid solidification. This process, characterized by a high solidification rate, can offer several advantages, such as the extension of solid solubility, precipitation of non-equilibrium or metastable crystalline phases, microsegregation-free solidification, and formation of a cellular structure with non-equilibrium morphologies, in particular in the stainless steel alloys [120,121]. It is well documented that this complex thermal history and the solidification process, as explained earlier, determine the final microstructure and, consequently, the performance of the produced parts. For this reason, in the following section, the aim is to provide an overview of the microstructure, residual stress, and surface quality of metallic materials processed via the LP-DED process.

### 4.1. Microstructure

In the LP-DED process, morphology and grain size as the main microstructural features can be significantly influenced by the thermal history of the component during the building process. In general, it has been reported that the cooling rate and thermal gradient can substantially define the thermal history of a part. However, it is well documented that there are lots of process parameters/variables that significantly affect the thermal history of the LP-DED parts. Therefore, this complex correlation between the process variables/parameters and thermal history makes predicting the microstructural features of LP-DED components very challenging. Nonetheless, in order to produce LP-DED parts with superior mechanical properties, it is very important to overcome this challenge and establish effective control mechanisms.

In the literature, several works have studied the role of specific parameters on the microstructure and performance of the LP-DED parts with specific shapes [121–123]. However, in most of these works, the investigations have been carried out on samples with simple geometries, such as cubes and blocks. In contrast, in the case of complex shape components that can experience various thermal histories in different parts, it is unclear how to apply the outcomes of these investigations. In the literature, it is reported that the as-built microstructure depends mainly on the solidification rate within the melt pool ($R$) and the thermal gradient at the solidification front ($G$) [124,125]. $G/R$ and $G \times R$, which is also known as cooling rate, are the two critical solidification parameters that influence the shape of the solidification front and the dimensions of the microstructure, respectively [126,127]. Columnar (elongated grain morphology), columnar-equiaxed, and equiaxed are the main structure morphology that can be achieved at different $G$ and $R$ values. Figure 24 shows the effects of $G$ and $R$ values on the solidified microstructure. As can be seen, the columnar to equiaxed transition can be achieved at very high solidification rates, whereas, at high cooling rates, a finer microstructure can be obtained in the as-built LP-DED parts. It should be emphasized that the optimum $G$ and $R$ values can be determined by different factors such as material properties, machine conditions, part geometries, and process parameters.

In general, in the LP-DED process, the cooling rate value ranges from $5 \times 10^2$ to $5 \times 10^5$ K/s [20,22,128]. The high cooling rate values, as discussed earlier, could lead to several advantages, such as suppression of solid-state transformation, formation of non-equilibrium phases, and very fine microstructures [20]. The $G \times R$ value at the bottom of the deposited track is very low, then increases rapidly with the track height until it reaches its maximum value, which is approximately equal to the travel speed near the free surface of the melt pool. On the other hand, the maximum value of $G/R$ is obtained at the bottom of the melt pool and decreases as it approaches the free surface of the melt pool. As a result, a coarse, planar microstructure is frequently obtained at the bottom of the deposited track, which evolves into cellular, dendritic or equiaxed structures in the upper regions. The

distribution of the microstructure is visible in Figure 25, which shows the microstructural evolution that occurs during the deposition of a single track of Co-based alloy on steel.

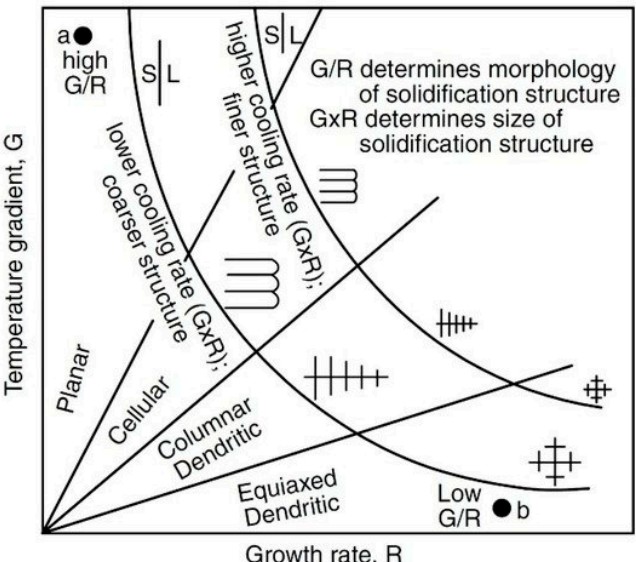

**Figure 24.** Effect of the temperature gradient and the solidification rate on the solidification microstructures and grain dimension [127].

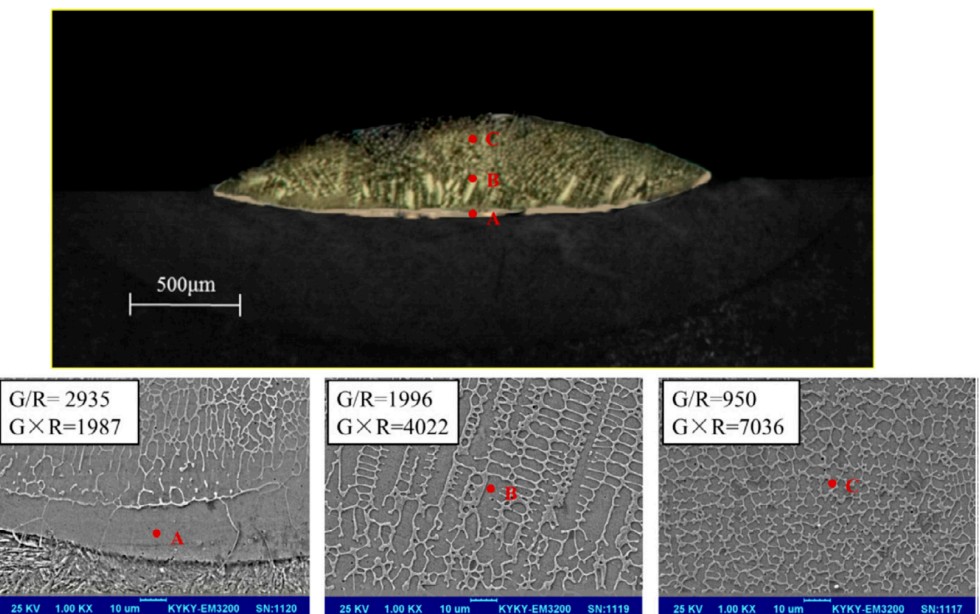

**Figure 25.** Experimental analysis of microstructure formation during the deposition of a single scan of Co-based alloy on steel. At the bottom of the deposited track (point A), a coarse planar front is observed. In the centre of the track (point B), columnar and coarse dendrite structures are observed. At the top of the track (point C), equiaxed and finer dendrites are obtained (reproduced with permission from [129]; Copyright © 2017 Elsevier).

Previous works have reported that epitaxial grain growth in the direction of maximum thermal gradient is the common mechanism in most metallic materials, such as stainless steel and titanium alloys [130,131], etc. However, various prior beta grain morphologies have been revealed in different LP-DED titanium components. For example, in a near beta titanium alloy (Ti-5Al-5Mo-4V-1Cr-1Fe), a "bamboo-like" grain morphology was detected

in the building direction, which is a mixture of small columnar and equiaxed grains, as shown in Figure 26 [132].

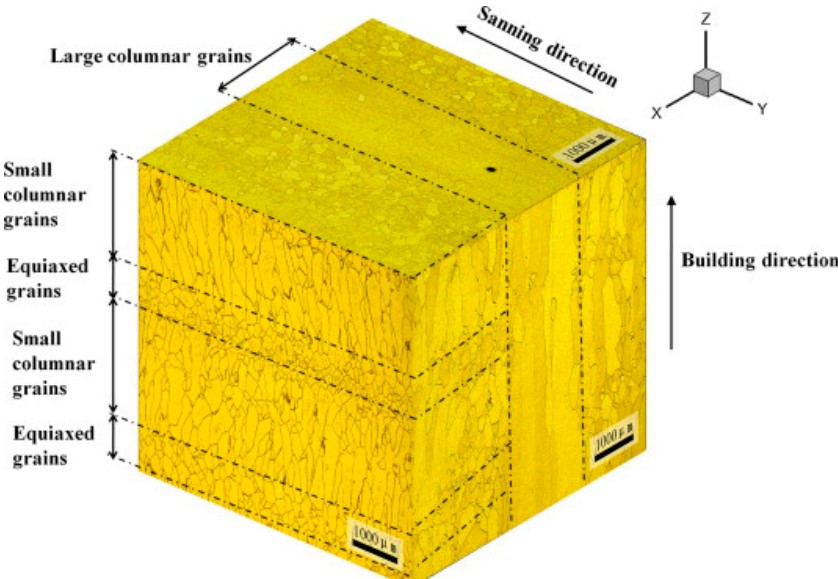

**Figure 26.** The "bamboo-like" grain morphology in an LP-DED Ti-5Al-5Mo-5V-1Cr-1Fe alloy (reproduced with permission from [132]; Copyright © 2013 Elsevier).

Liu, et al. [132] studied the correlation between the microstructural evolution of a near beta titanium alloy during the LP-DED process and the part geometry (Figure 27).

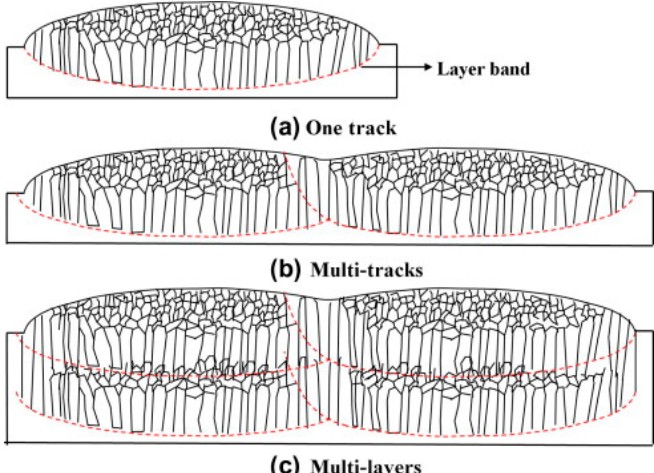

**Figure 27.** Schematic illustration of the microstructural evolution during the LP-DED process, (**a**) One-track, (**b**) Multi-track, and (**c**) Multi-layers (reproduced with permission from [132]; Copyright © 2013 Elsevier).

As seen in Figure 27a, during the deposition of a single track, due to the high-temperature gradient and rapid cooling at the bottom of the melt pool, the morphology of the grains is columnar. Thereafter, as the height of the melt pool increases, the thermal gradient and cooling rate decrease, resulting in a transition from columnar to equiaxed at the top of the melt pool. Instead, in the case of multi-track deposition (Figure 27b), in the overlapping zones, all the equiaxed grains remelt and subsequently solidified with a columnar morphology. This transition in these areas is due to the shallower melt pool depth in these zones, which changes the local thermal history in those locations. This different thermal history, particularly higher thermal gradients and cooling rates, leads to

the formation of columnar grain. In the case of multi-layer deposition, shown in Figure 27c, it is revealed that partially melted equiaxed gains of the last layer solidify and act as nuclei for the epitaxial growth of the columnar grains, which have an identical morphology to that of the previous layer. The grain morphology in the overlapped zones, in this case, is also the same as in multi-track deposition. However, they revealed that a variation in the LP-DED process parameters could change grain morphologies, so further studies on the effect of process parameters on microstructural evolution are needed.

In another work, Wang, et al. [133] studied the effect of mass deposition rate ($m$) on the nucleation and growth mechanisms during the solidification of the melt pool in the LP-DED process. They reported that by increasing the mass deposition rate, the area fraction of the equiaxed grains increases significantly (Figure 28). In fact, by increasing the $m$ value, the epitaxial growth of the parent grains is more prevented, and nucleation of the new equiaxed grains increases significantly. Besides, it was revealed that higher mass deposition rates result in a lower melting pool depth, and after a critical point, no penetration occurs, which is undesirable. Therefore, they concluded that through the mass deposition rate, it would be possible to change the morphology of the microstructure, but there is a critical threshold that should be respected.

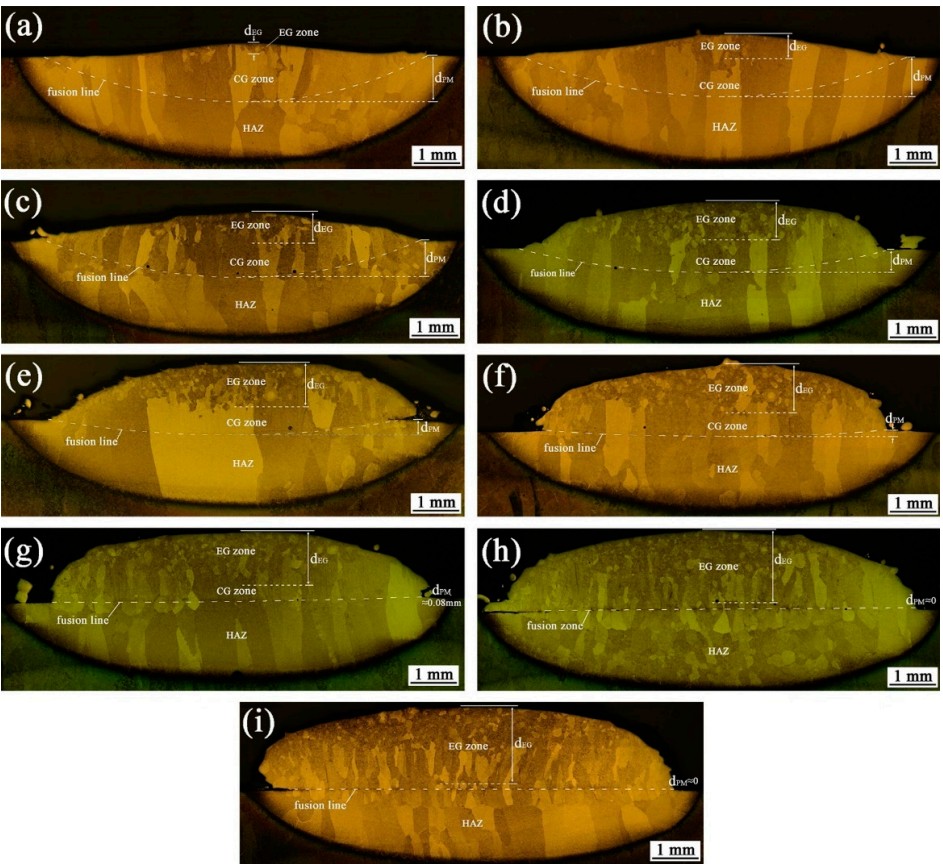

**Figure 28.** OM images of the cross-section of melt pools of a near beta titanium single scan tracks using (**a**) 6 g/min, (**b**) 11 g/min, (**c**) 15 g/min, (**d**) 25 g/min, (**e**) 31 g/min, (**f**) 36 g/min, (**g**) 44 g/min, (**h**) 55 g/min and (**i**) 59 g/min, as mass deposition rate (reproduced with permission from [133]; Copyright © 2015 Elsevier).

Hofmeister, et al. [93] analyzed the microstructure of 316L stainless steel thin walls. By varying the laser power, they found that a columnar microstructure was predominant near the interface, whereas a cellular microstructure was observed in the remaining regions. They observed that the dimensions of the microstructures were smaller using a low value of laser power. Mazumder, et al. [128] analyzed the microstructure obtained from the deposition of

H13 tool steel. They observed that each deposition pass was characterized by three regions: the interface region, the columnar grain, and the equiaxed grain region (Figure 29). The interface region corresponds to the remelted region and is characterized by a coarser mesh due to repeated thermal cycles. Above the interface region, the microstructures evolve into the columnar grain and then into the equiaxed grain, which has been attributed to a lower value of temperature gradient [128].

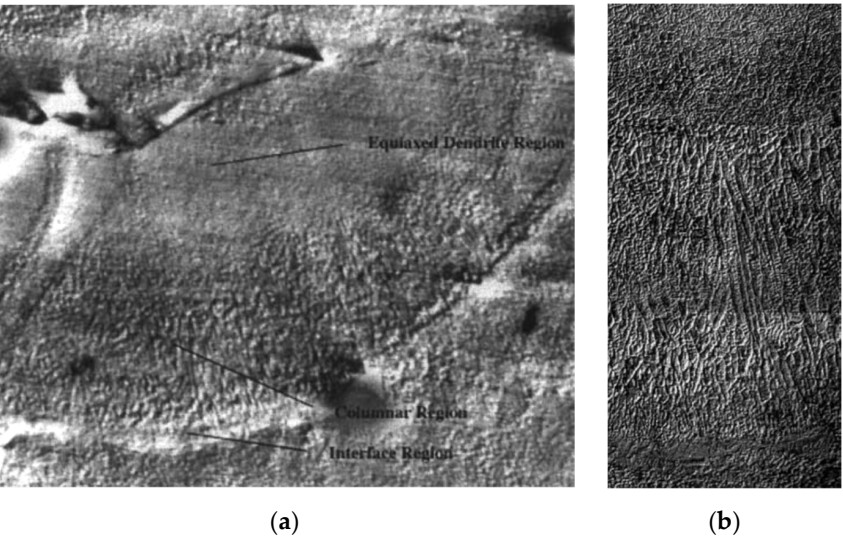

(**a**)　　　　　　　　　　　　　　　　(**b**)

**Figure 29.** (**a**) Cross-section of H13 tool steel deposit and (**b**) close-up of deposition pass characterized by three-zone: interface region, columnar region, and equiaxed dendrite region (reproduced with permission from [128]; Copyright © 1999 Springer Nature).

Mazumder, et al. [128] and Han, et al. [134] demonstrated that the microstructures are highly influenced by the process parameters. In particular, increasing the laser power and\or reducing the travel speed result in coarser grains. The same behavior was observed by Wu, et al. [135] for Ti-6Al-4V alloy and by Majumdar, et al. [136] for 316L stainless steel. Figure 30 shows the variation of grain structure dimensions obtained during the deposition of Ti-6Al-4V thin walls by increasing the travel speed. Liu, et al. [137] deposited Inconel 718 samples on a substrate of the same material and observed that the microstructure was mainly characterized by elongated columnar dendrites along the building direction. They also observed that a finer grain structure was obtained in the regions of overlap between consecutive tracks, while a coarser grain structure was obtained in the other regions (Figure 31). This was attributed to the thermal cycles that occur during the process, which in turn cause the grain recrystallization and coarsening.

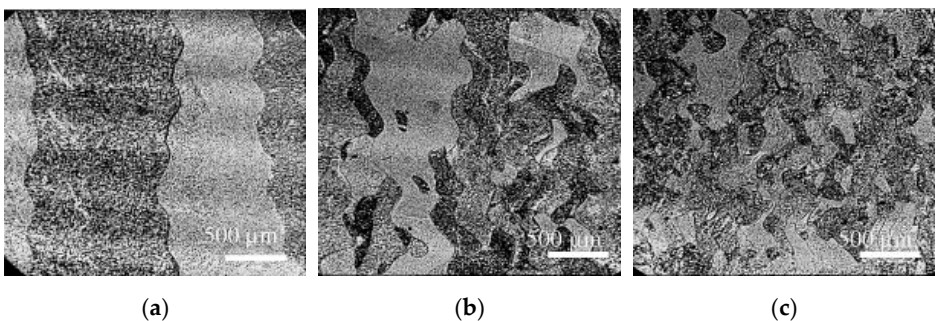

(**a**)　　　　　　　　　　　(**b**)　　　　　　　　　　　(**c**)

**Figure 30.** SEM micrograph showing the effect of travel speed on grain dimensions. Travel speeds are (**a**) 300 mm/min, (**b**) 600 mm/min, and (**c**) 900 mm/min (reproduced with permission from [135]; Copyright © 2004 Elsevier).

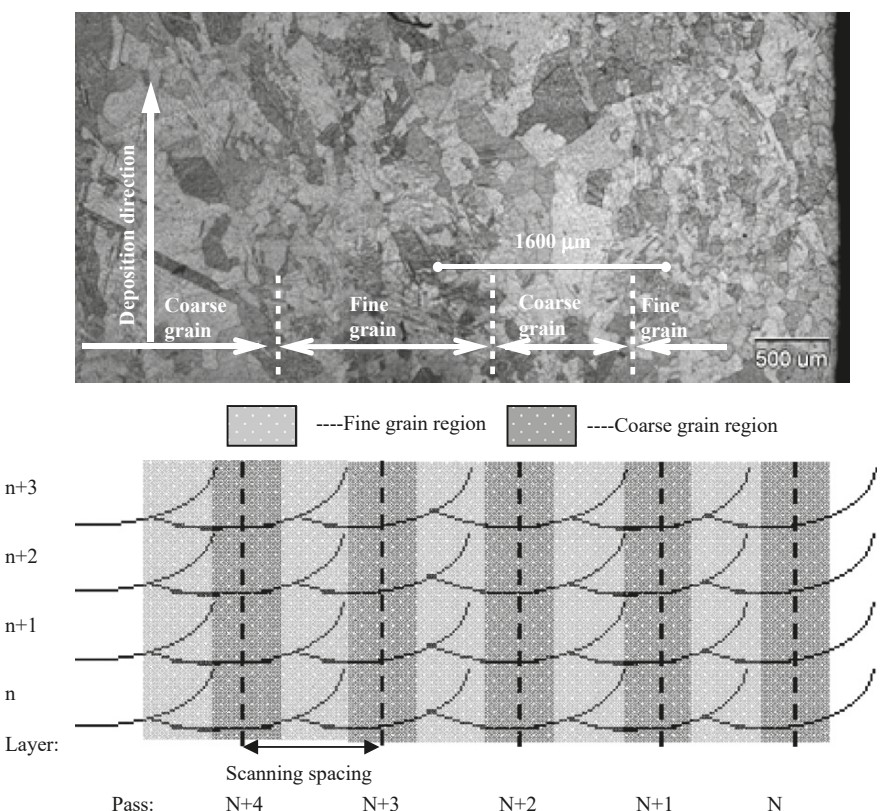

**Figure 31.** Cross-section of In718 deposit. Finer grain structure is observed in the overlapping regions, and coarser grain structure is observed elsewhere (reproduced with permission from [137]; Copyright © 2011 Elsevier).

Several works have illustrated that in a metal part produced via the LP-DED process, different local thermal gradients and cooling rates can lead to the formation of various local microstructures [120,138]. For instance, in a Ti-6Al-4V component produced via LP-DED, the colonies of parallel and very fine lamellae are formed at the top of the sample (Figure 32); in contrast, at the bottom of the sample, the lamellae become thicker [130].

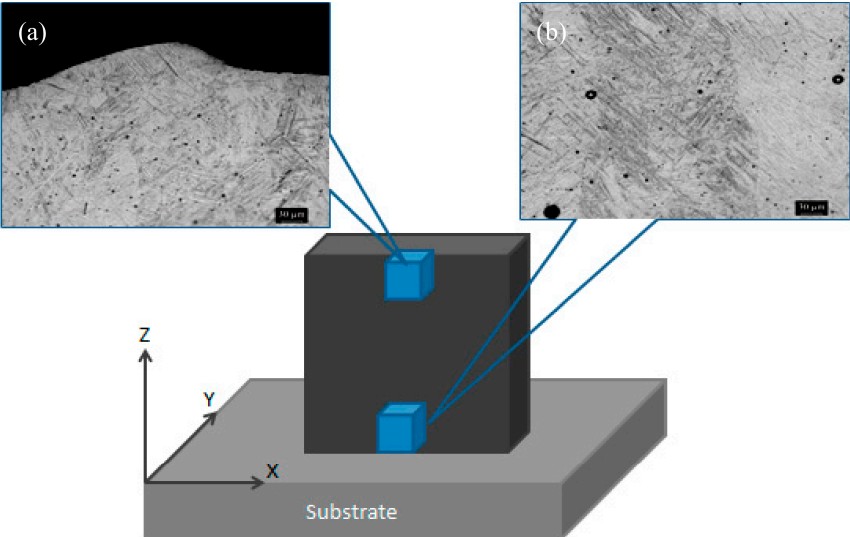

**Figure 32.** OM micrographs of a Ti-6Al-4V sample produced via LP-DED process from the (**a**) top to the (**b**) bottom region [130].

*4.2. Residual Stress*

The dynamic nature of thermal phenomena in LP-DED processes induces a high level of residual stresses. These residual stresses develop due to the reiteration of heating/cooling cycles and the high-temperature gradient during the process. According to Mercelis and Kruth [139], two mechanisms are primarily responsible for the generation of residual stresses: the Temperature Gradient Mechanism (TGM) and the cooling-down phase. The TGM is caused by the high-temperature gradient near the laser spot, whereas in the cooling-down phase model, the molten material shrinks during the solidification due to thermal contraction. However, this contraction is limited by the underneath material [139–141]. The magnitude of residual stresses is influenced by the stress-strain relationship of the material and by the strain misfit between two adjacent regions during the cooling phase [142]. In particular, higher residual stresses have been observed in materials characterized by a higher modulus of elasticity ($E$) and higher yield strength ($Ys$). For example, Rangaswamy, et al. [143] showed that residual stresses in the IN718 sample ($E$ = 205 GPa, $Ys$ = 1100 MPa) were approximately 1.5 times higher than residual stresses in the 316L sample ($E$ = 196 GPa, $Ys$ = 450 MPa). Residual stresses in some areas can reach very high values, even 75% of the yield strength [143].

The strain misfit between adjacent regions is crucial during the production of Functionally Graded Material (FGM), where materials with very different thermal expansion coefficients (CTE) are used. Woo, et al. [144] studied the residual stress distribution obtained in five FGM samples produced with different chemical compositions. They observed that as the chemical composition of the samples changed, there was an abrupt variation from tension to compression.

The presence of residual stresses affects important characteristics of the produced components, such as tensile and fatigue strength, thus influencing component integrity and service life [138]. Moreover, the residual stresses induce distortion [145,146] and loss of geometrical tolerances [147]. Due to complex heating and cooling cycles, stresses in the deposited material assume a highly non-uniform distribution. Rangaswamy, et al. [148] measured the residual stresses of two stainless steel samples using the neutron diffraction method. They analyzed samples with different geometries, i.e., a thin wall and a pillar with a square cross-section. They showed that a compression state is observed at the centre of each sample, while a tension state is obtained near the edge. Later, Rangaswamy, et al. [143] measured the residual stress distribution of two samples with rectangular and square cross-sections using neutron diffraction and the contour method. They observed that the stresses were almost uniaxial along the building direction. Moat, et al. [149] analyzed the stress distribution on parallelepiped samples using the contour method. The results, illustrated in Figure 33, showed that near the building platform, the stresses along the building direction were almost zero; the longitudinal stresses, on the other hand, were highly compressive. On the contrary, the stress distribution on top layers showed tensile stresses along the building direction, but longitudinal stresses had almost zero values. This behavior was observed by analyzing the macroscale residual stress distribution and confirmed by several authors [150–154]. Nevertheless, the high non-uniformity of residual stresses can also be observed in the mesoscale and the microscale levels, which analyze the stress distribution on the layer track length scale, respectively [137,155,156]. Strantza, et al. [155] measured the residual stresses at the mesoscale level of Ti-6Al-4V samples produced by LP-DED using hole drilling and slitting methods. They proved that the samples were characterized by a tensile state on the external surface that was balanced by a compression state in the centre. High levels of residual stresses were observed near the bottom due to the high value of the cooling rate and the cumulative effect of successive depositions. In particular, using the hole-drilling strain gauge method, it was observed that the residual stresses are characterized by an oscillatory nature, in which the amplitude of the oscillation is correlated to the layer thickness and the hatching distance [155,157]. Using the Vickers micro-indentation method on Inconel 718 samples, Liu, et al. [137] showed that higher stresses are obtained in the overlapping regions between two adjacent tracks.

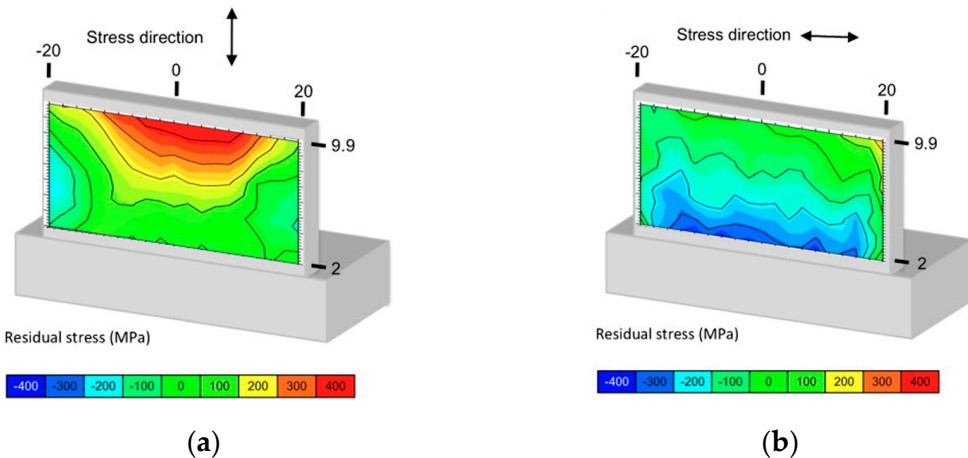

**Figure 33.** Residual stress distribution in the sample (**a**) along the building direction and (**b**) along the longitudinal direction (reproduced with permission from [149]; Copyright © 2011 Elsevier).

Since residual stresses can adversely affect the mechanical performance and functionality of the part, several efforts have been made to define methods to minimize residual stresses. To this end, three main aspects were investigated: building platform temperature, deposition strategy, and process parameters.

Based on the results gained from welding processes, Chin, et al. [158] simulated the deposition of a single steel track. They showed that preheating the building platform caused a reduction of the thermal gradient and residual stresses. Vasinonta, et al. [159] proposed a thermo-mechanical model and developed a process map to predict residual stresses. They showed that the main contribution to the reduction of residual stresses came from the uniform preheating of the building platform. Corbin, et al. [160] used a laser sensor to analyze the effect of preheating temperature and building platform thickness on deformation during the deposition of Ti6Al4V alloy. They showed that the preheating phase reduced distortion when a thin building platform is used; however, when a thick building platform is used, the preheating phase induces more deformation. Lu, et al. [145] used a Finite Element (FE) to simulate the deposition of Ti64Al4V samples. They analyzed the influence of different laser paths used to increase the temperature of the building platform locally. They showed that when local preheating was used, the residual stress value was lower than in the nominal case without preheating. However, the distortion increased by approximately 3 mm. Instead, by increasing the temperature of the entire building platform to 700 °C, the residual stresses and distortions of the Ti64Al4V samples were reduced by 80.2% and 90.1%, respectively, compared with the nominal case.

Optimization of the deposition strategy is another method used to achieve a reduction in residual stresses. Saboori, et al. [121] compared two deposition strategies: the 0–90°, characterized by an orthogonal deposition direction between two layers, and the 0–67°, which is characterized by a 67° rotation for each new layer. They showed that the residual stresses on the top surfaces were independent of the deposition strategy; however, higher residual stresses were observed on lateral surfaces produced by the 0–90° deposition strategy. Woo, et al. [144] measured residual stresses in FGM samples produced with different deposition strategies. They showed that the range of stress, i.e., the difference between the maximum and the minimum value, was reduced from 950 MPa to 680 MPa by rotating the deposition strategy by 90° at each layer. The lower range value of about 430 MPa was obtained using an island deposition strategy. However, using the island deposition strategy, a higher level of defects was observed. Dai and Shaw [161] and Nickel, et al. [162], using a FE model analysis, showed that residual stresses and related distortions depend significantly on the laser deposition strategy. Using a bi-directional deposition strategy elongated along a direction, the resulting distortion was characterized by a saddle shape (Figure 34a,c). The distortion was reduced by varying the laser deposition pattern.

In fact, by using an offset-out deposition strategy (Figure 34b,d), the induced distortion was reduced to about 1/3 compared to that obtained with the first deposition pattern [161].

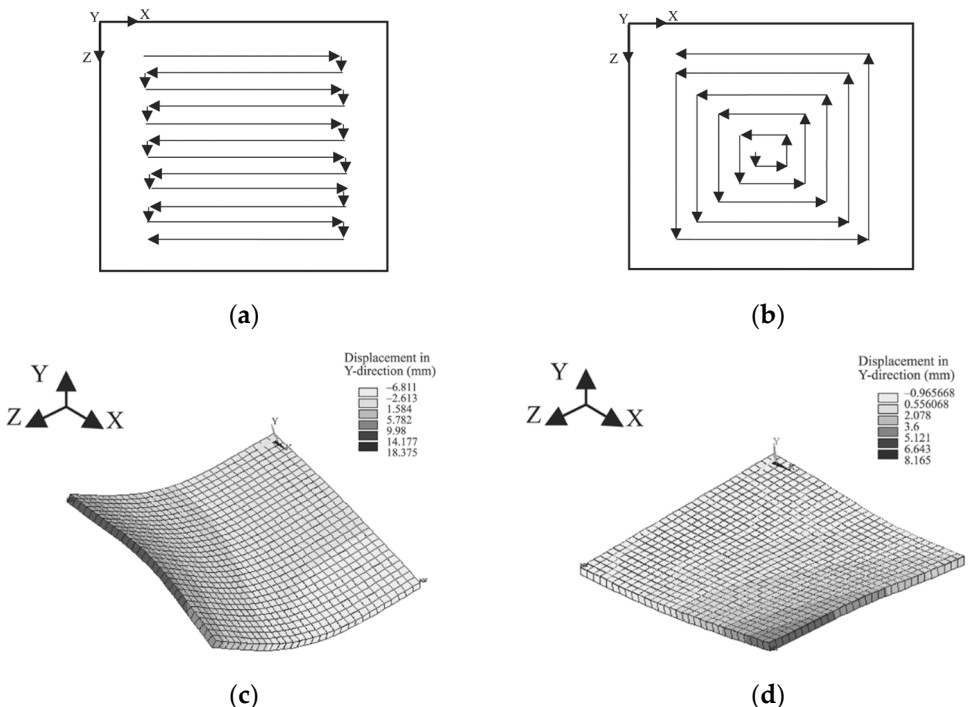

**Figure 34.** (**a**) Bi-directional and (**b**) offset-out deposition strategies and (**c**,**d**) the corresponding modeled deformation caused by residual stress (reproduced with permission from [161]; Copyright © 2002 Emerald Publishing Limited).

Mitigation of residual stresses can also be achieved by varying the process parameters. In particular, lower residual stresses are obtained by varying the process parameters in order to obtain a constant melt pool dimension and a uniform temperature distribution throughout the component [143,163]. In the literature, contradictory results are found concerning the effect of process parameters on residual stress values. All in all, it is observed that an increase in specific energy, either through an increase in laser power or a reduction in travel speed, leads to an increase in residual stress values [163–166]. Then, Balichakra, et al. [167], during the deposition of a γ titanium aluminide thin wall, showed that residual stresses were almost independent of the travel speed and laser power. On the other hand, another author showed that residual stresses were lower when using higher laser power [168] and lower travel speed [165,168,169].

*4.3. Surface Quality*

Components produced by the LP-DED process suffer from low surface quality. Hence, post-machining operations are required in order to achieve the dimensional and geometrical tolerances defined by the specific application. The surface quality of a component is mainly characterized by its surface roughness and dimensional accuracy [170].

Surface roughness is one of the most challenging issues in LP-DED [20,171,172]. Smugeresky, et al. [173] performed one of the first experimental investigations on the surface finish of samples produced by the LP-DED process. Their analysis showed that the average surface roughness ($R_a$) ranges between 8 and 20 μm, and the surface roughness value is mainly influenced by powder particle size. In particular, the lowest value of surface roughness was obtained when the smallest powder particle size was used. This behavior was attributed to the presence of unmelted powder particles on the analyzed surface [171]. In addition to the particle size, the literature shows that the surface roughness is highly influenced by the powder stream [174].

Based on the powder flow rate value, Resch, et al. [175] distinguished the surface roughness into three categories, which are related to the low, medium, and high powder flow rate values. In detail, using a low value of powder flow rate resulted in smoother surfaces and the lowest values of surface roughness, but the build rate was very low. Using a higher value of powder flow rate, the build rate increased, but the surface roughness increased significantly. Using a medium value of powder flow rate, a good building rate with reasonable surface roughness was observed. The surface roughness is also influenced by other process parameters such as layer thickness, laser power, and travel speed.

Gharbi, et al. [176] noticed that if a thin additive layer is used in combination with high laser power and high travel speed, an improvement in the surface finish is observed. This was confirmed by Peyre, et al. [171], that showed that increasing the laser power results in a reduction of the maximum surface roughness value; however, the average surface roughness remains almost constant. In contrast, Mahamood, et al. [177] and Li, et al. [172] observed that average surface roughness decreases if the laser power value increases. Gharbi, et al. [176] identified two types of roughness along the building direction (z-axis): micro-roughness and macro-waviness. The former was attributed to the particle agglomeration in inter-layer areas and the solidification line and was mainly influenced by the powder stream, while the macroscopic contribution, which was mainly influenced by the process parameters, was related to the formation of periodic menisci, associated with the stability of the melt pool. In general, surface roughness can be measured on the top surface (building plane) and the lateral surfaces (along the building direction) [178].

Despite the fact that most works have analyzed the surface roughness on the lateral surfaces [173,175,176,179], it is observed that the surface roughness on the top surfaces is slightly higher than that measured on the lateral surfaces [168,180]. In particular, Piscopo, et al. [168] showed that the morphology of the lateral and the top surface is mainly influenced by the layer thickness and the hatching distance, respectively. In addition, by analyzing the morphology of the surface described by the kurtosis and the skewness parameters, they showed that the morphology of an LP-DED sample is similar to that obtained in a milling operation. In addition to varying process parameters, different methods have been developed to improve surface roughness. Gharbi, et al. [176] showed that the surface finish is improved using quasi-continuous laser irradiation instead of fully continuous laser irradiation. This was attributed to the reduction of the thermal gradient and Marangoni flow in the melt pool.

One of the main methods used to reduce top surface roughness is surface scanning without powder feeding. This method, in analogy with the L-PBF process, is named remelting. Rombouts, et al. [181] studied the effect of laser remelting on surface roughness. Figure 35 illustrates the surface profile data before and after remelting. In their work, the surface quality was evaluated in terms of maximum profile height (Rt) along two directions parallel (X-direction) and perpendicular (Y-direction) to the deposition path. Before remelting, a lower value of Rt was observed in the analyzed samples. Moreover, a clear correspondence between waviness and hatching distance was identified on the deposited samples. This correspondence was not observed after remelting.

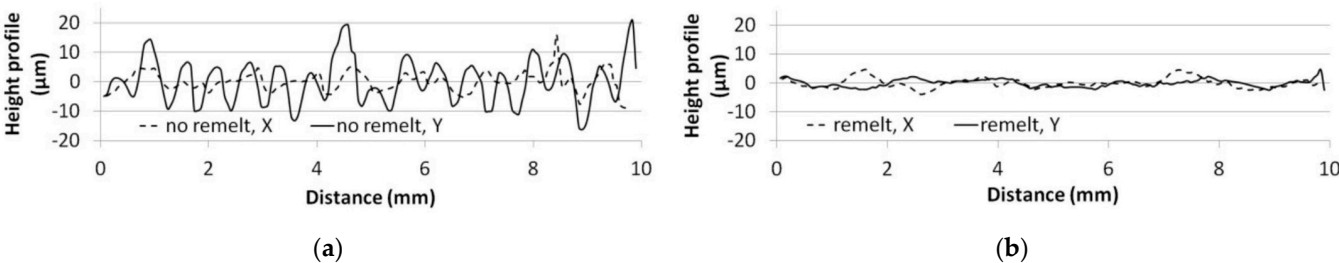

(**a**)            (**b**)

**Figure 35.** Height profile measured on samples (**a**) without and (**b**) with laser remelting (reproduced with permission from [181]; Copyright © 2013 Elsevier).

Dimensional accuracy is another important factor used to describe surface quality. In fact, based on dimensional and geometrical deviation, the post-processing operation can be designed [168]. Izadi, et al. [182] and Piscopo, et al. [168] reported that dimensional deviation depends on process parameters. Gruber, et al. [183], based on the dimensional results, showed that the accuracy of the LP-DED process falls into the coarse tolerance class. However, the accuracy can be increased by optimizing the process parameters [168,183]. For example, Piscopo, et al. [168] observed that travel speed significantly influences the dimensional deviation, which decreases as the travel speed increases. On the other hand, the dimensional deviation was not significantly influenced by the laser power.

Table 5 summarizes the main aspects of the solidification process and its most important parameters.

**Table 5.** Summary of the factors influencing the solidification mechanisms.

|  | Microstructure | Residual Stress | Surface Quality |
|---|---|---|---|
|  | Control the morphology and the dimension of the grains | Influence the internal stress state and the part/substrate distortions | Influence the surface roughness and the dimensional accuracy |
| Material properties | [130–132,137] | [143,144,148] |  |
| Thermal gradient and solidification rate | [124–127] | [150–154] |  |
| Laser power | [93,128,134] | [143,145,168] | [171,176] |
| Travel speed | [128,134,135] | [143,163] | [171] |
| Powder flow rate | [133] |  | [174,175] |
| Deposition strategy | [128] | [121,137,144,161,162] | [168,181] |
| Substrate temperature |  | [158–160] |  |

## 5. Conclusions

The purpose of this review is to provide a summary and analyse the main mechanisms involved in the Laser Powder Directed Energy Deposition process (LP-DED). From the literature review, three mechanisms are identified: laser irradiation and material addition, the generation of the melt pool, and the solidification. For each of these mechanisms, the most significant parameters and their effect on the deposition characteristic have been discussed. Compared to the currently available literature, which focuses on specific aspects of the process, this paper provides a comprehensive overview of the main mechanisms of the LP-DED process.

The process analysis highlighted the complexity of the phenomena occurring during the deposition process and suggested future studies are needed to improve the process. Some possible activities should focus on the development and testing of new deposition heads that can maximize capture efficiency, the analysis of the effect of a wider range of process parameters and a wider range of materials, the investigation of new deposition strategies and process parameters for residual stress mitigation, and, finally, the structured analysis on the dimensional and geometrical capabilities of the LP-DED process.

**Author Contributions:** Conceptualization, G.P., A.S. (Alessandro Salmi), A.S. (Abdollah Saboori) and E.A.; writing—original draft preparation, G.P., A.S. (Alessandro Salmi), A.S. (Abdollah Saboori) and E.A.; writing—review and editing, G.P., A.S. (Alessandro Salmi), A.S. (Abdollah Saboori) and E.A. All authors have read and agreed to the published version of the manuscript.

**Funding:** This research received no external funding.

**Institutional Review Board Statement:** Not applicable.

**Informed Consent Statement:** Not applicable.

**Data Availability Statement:** Not applicable.

**Conflicts of Interest:** The authors declare no conflict of interest.

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
