# Peer review of "An Overview of the Process Mechanisms in the Laser Powder Directed Energy Deposition"

_applsci, doi:10.3390/app13010117_

Round 1

Reviewer 1 Report

this paper summarizes the whole process character of the Laser Powder Directed Energy Deposition (LP-DED) including : powder material, powder flow, the generation of the molten pool and the solidification. it is a meaningful papers to understand this technical. the Englis also write freely and organiazed well. so we suggest it can be accept.

Reviewer 2 Report

In this manuscript, three mechanisms involved in the LP-DED process (powder flow, melt pool generation, solidification process) has been summarized. The highly developed logical structure has been showed in this manuscript. However, there are still some shortcomings that need further revision. The details were shown as follows.

(1)In introduction, the application background of AM needs to be further stated, e.g. the area of application (use) of AM and what advantages it has compared to similar technologies?

(2)The serial number of Fig. 4 is wrong, please check whether the order of (a), (b), (c) corresponds to the figures; please check similar issues in detail and revise them together.

(3)Section 2.1 only states the results of previous research and the author lacks a summary of these results. It is worth noting that this section, as one of the key components of the paper, needs to be further summarisd and distilled, then present the valuable and correct results in this section, rather than writing a literature review like introduction.

(4)in line 445~448, the authors cite two different relationships between layer height and laser power obtained by Srivastava and Lee et al. respectively. However, as a review paper, the author should give his or her own scientific and accurate judgement. It is more important to summarise accurate conclusions and provide guidance for future research, rather than just listing, which will have few guidance for the industry.

(5)The conclusion needs further revision and the research significance of this paper need to be added.

(6)there are some formatting problems and missing information in references. The authors are requested to double-check and make corrections. The author needs to double-check the missing parts and make corrections.

Reviewer 3 Report

The reviewed manuscript presents a review on one of the most significant additive manufacturing processes which is Laser Powder Directed Energy Deposition (LP-DED). The authors mentioned the advantage of such technique for different application and the claimed that the main shortage is the knowledge related to the physics of the process mechanisms. Consequently, the authors tried to make a review that cover the main mechanisms of such technique besides the relationship between the process parameters and the outcome of each step of the process. I have found the paper to be interesting. However, some concerns need to be addressed before accepting the paper for publication to improve the readability and clarity of the manuscript:

1-    The title doesn’t reflect the importance of the review. I think the title should be modified to clarify the significance of the review. In my opinion the title could be modified to be “An Overview of the Laser Powder Directed Energy Deposition Process Main Mechanisms”.

2-    Please consider reviewing the abstract and highlighting the novelty. I suggest reorganizing the abstract, highlighting the novelties introduced. The abstract should contain answers to some questions, what problem was studied and why is it important? and what conclusions can be drawn from the results? (Please provide generic ones and not specific results, as in review papers abstract should be descriptive rather than informative).

3-    The conclusion section is too long. Conclusions should be concise to illustrate the important outputs of the research. Furthermore, where are the recommendations for future work to help other researchers open new windows toward the state of the art in this field.

4-    The introduction needs some enhancement, by adding some updated references in 2022. Although we are at the end of 2022, the author only mentioned 3 papers in 2022 among 170 references which illustrates the oldness of the topic and the researchers reluctance to work on such a topic. Many papers in 2022 have considered the the utilization of Laser Powder Directed Energy Deposition:

·      Piscopo, Gabriele, and Luca Iuliano. "Current research and industrial application of laser powder directed energy deposition." The International Journal of Advanced Manufacturing Technology (2022): 1-25.

·      Wu, Jiahzu, et al. "Modeling of whole-phase heat transport in laser-based directed energy deposition with multichannel coaxial powder feeding." Additive Manufacturing 59 (2022): 103161.

·      Wang, Jin, et al. "Evaluation of in-situ alloyed Inconel 625 from elemental powders by laser directed energy deposition." Materials Science and Engineering: A 830 (2022): 142296.

·      Svetlizky, David, et al. "The influence of laser directed energy deposition (DED) processing parameters for Al5083 studied by central composite design." Journal of Materials Research and Technology 17 (2022): 3157-3171.

5-    When discussing and comparing results, it better to conclude results in tables to make the comparison clearer.

6-      There are a lot of sentences which either do not make sense or lacking some additional words to complete their meaning. A thorough review of the article content is needed to improve the quality of the text.

7-      To illustrate the importance of the paper, the authors need to clarify the difference between the approach covered in the current review and other approaches. The accuracy of each approach?

8-      The authors should make sure that the format of tables and figures is consistent (i.e. same size, style, format..etc).

9-      The use of English language is reasonable, however, there are a number of punctuation and grammatical errors; that should be corrected and rephrased using academic English for a better flow of text for reader.

Please, read the text carefully before the next submission of the paper.

Round 2

Reviewer 3 Report

Many thanks for the revision and for incorporating all suggested changes to the manuscript that are nicely reflected. The authors did a good job of improving the article. I believe that the article has become much better, and now I recommend this article for publication.